

# A Maxwell–Elasto-Brittle rheology for sea ice modelling

**V. Dansereau[1], J. Weiss[2], P. Saramito[3], and P. Lattes[4]**

[1]Laboratoire de Glaciologie et Géophysique de l'Environnement, CNRS – UMR5183,
Université de Grenoble, Grenoble, France
[2]Institut des Sciences de la Terre, CNRS – UMR5275, Université de Grenoble,
Grenoble, France
[3]Laboratoire Jean Kuntzmann, CNRS – UMR5224, Université de Grenoble, Grenoble, France
[4]TOTAL S. A. – DGEP/DEV/TEC/GEO, Paris, France

Received: 1 November 2015 – Accepted: 30 November 2015 – Published: 15 January 2016

Correspondence to: V. Dansereau (veronique.dansereau@lgge.obs.ujf-grenoble.fr)

Published by Copernicus Publications on behalf of the European Geosciences Union.

**TCD**

doi:10.5194/tc-2015-200

A Maxwell–Elasto-Brittle rheology for sea ice modelling

V. Dansereau et al.

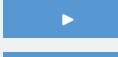

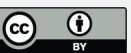

## Abstract

A new rheological model is developed that builds on an elasto-brittle (EB) framework used for sea ice and rock mechanics, with the intent of representing both the small elastic deformations associated with fracturing processes and the larger deformations
occurring along the faults/leads once the material is highly damaged and fragmented. A viscous-like relaxation term is added to the linear-elastic constitutive relationship together with an effective viscosity that evolves according to the local level of damage of the material, like its elastic modulus. The coupling between the level of damage and both mechanical parameters is such that within an undamaged ice cover the viscosity is infinitely large and deformations are strictly elastic, while along highly damaged zones the elastic modulus vanishes and most of the stress is dissipated through permanent deformations. A healing mechanism is also introduced, counterbalancing the effects of damaging over large time scales. In this new model, named Maxwell–EB after the Maxwell rheology, the irreversible and reversible deformations are solved for simultaneously, hence drift velocities are defined naturally. First idealized simulations without advection show that the model reproduces the main characteristics of sea ice mechanics and deformation: strain localization, anisotropy, intermittency and associated scaling laws.

## 1 Introduction

Making reliable predictions of the drift and deformation of sea ice is becoming crucial nowadays for: (1) forecasting the opening of shipping routes across the Arctic, (2) evaluating mechanical constraints on offshore structures and ships and, at larger scales, (3) estimating the future evolution of both the summer and winter sea ice cover in the Arctic and Antarctic to anticipate its short to long-term, regional to global impacts on climate. Current operational modelling platforms, whether assimilating data of not (e.g. TOPAZ4: Sakov et al., 2012; GIOPS: Smith et al., 2015), and global climate

**TCD**

doi:10.5194/tc-2015-200

**A Maxwell–Elasto-Brittle rheology for sea ice modelling**

V. Dansereau et al.

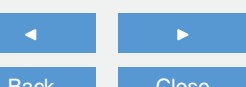

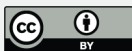

models including sea ice dynamics (e.g. the Coupled Model Intercomparison Project Phase 5 models involved in the IPCC Fifth Assessment Report, Flato et al., 2013) are based on the same mechanical framework for sea ice developed in the late seventies: the Hibler Viscous-Plastic (VP) model (Hibler, 1977, 1979). With this approach, the ice creeps very slowly as a viscous fluid under small stresses and deforms plastically once exceeding a yield criterion. Yet, over the last decade, the viscous hypothesis and other underlying physical assumptions of this VP framework have been revisited and found inconsistent with the observed mechanical behaviour of sea ice (Weiss et al., 2007; Coon et al., 2007; Rampal et al., 2008). In the same line of ideas, recent modelling studies have demonstrated that while the VP model can represent with a certain level of accuracy the mean, global (> 100 km) drift of sea ice, it fails at reproducing the observed properties of sea ice deformation and that, especially at the fine scales (Lindsay et al., 2003; Kwok et al., 2008; Girard et al., 2009) relevant for operational modelling, thereby stressing the need to explore alternative rheologies.

Other continuum models have been developed lately with the aim of representing more accurately some important aspects of the mechanical behaviour of sea ice. Considering the discontinuous and anisotropic character of the pack, Schreyer et al. (2006) have suggested an elastic-decohesive model that explicitly accounts for the deformation arising from discontinuities in displacement across leads, the orientation of which is prescribed. Tsamados et al. (2013) have presented a model based on the rheology of Wilchinsky and Feltham (2006) that accounts for the subgrid scale anisotropy of the sea ice cover. Their framework incorporates an evolution equation for the orientation of ice floes, for which a diamond shape is assumed. Our present work shares the same objective as these previous initiatives: to build a continuum model for sea ice that is physically consistent with its observed mechanical behaviour. However, we chose to base our approach on a completely isotropic rheology and, by incorporating the relevant brittle mechanics concepts and long-range elastic interactions, aim to develop a model that reproduces the anisotropy and extreme gradients within the sea ice

**TCD**

doi:10.5194/tc-2015-200

**A Maxwell–Elasto-Brittle rheology for sea ice modelling**

V. Dansereau et al.

cover naturally, that is, without the need of treating velocity discontinuities explicitly nor prescribing lead orientations or floe shapes.

Early on, sea ice scientists have suspected that the sea ice cover behaves in a brittle instead of a viscous manner, with some strain hardening in compression (Nye, 1973).
Studies of fracture patterns, stresses and strains both in situ and in the laboratory have suggested that the deformation of sea ice is mostly accommodated by a mechanism of multiscale fracturing and frictional sliding (Marsan et al., 2004; Weiss et al., 2007; Schulson and Duval, 2009; Schulson, 2006). By investigating the dispersion of ice buoys, Rampal et al. (2008) recently showed that sea ice over the Arctic deforms in a heterogenous and intermittent manner over spatial scales of 300 m to 300 km and time scales of 3 h to 3 months. The strong space–time coupling in the scaling laws revealed by their analyses are consistent with (1) a brittle-type material in which permanent deformations are accommodated by displacements along fractures and fault planes over a wide range of scales and (2) long-range elastic interactions, allowing for small, local perturbations to trigger much larger damaging events within the ice pack (Marsan and Weiss, 2010).

A close comparison can be made between the deformation of sea ice and that of the Earth crust, in which brittle fracturing and Coulomb stress redistribution also take place and for which scaling properties have been recognized for years (Kagan and Knopoff, 1980; Kagan, 1991; Kagan and Jackson, 1991; King et al., 1994; Turcotte, 1992; Stein, 1999). Recently, Marsan and Weiss (2010) established a formal analogy between the mechanical behaviour of sea ice and that of the Earth crust by demonstrating that the space–time coupling in the deformation of sea ice, estimated from *continuous* displacement fields, is equivalent to a coupled scaling of the *discrete* ice-fracturing events occurring along the leads, similar to that observed for earthquakes (Kagan, 1991; Kagan and Jackson, 1991). The authors suggested that the similarity between sea ice and the Earth crust is attributable to a common cascading mechanism of earth-/ice-fracturing events that extends the influence of local events to longer durations and larger areas than their direct aftershocks.

Discussion Paper | Discussion Paper | Discussion Paper | Discussion Paper |

**TCD**

doi:10.5194/tc-2015-200

**A Maxwell–Elasto-Brittle rheology for sea ice modelling**

V. Dansereau et al.

In the case of rocks, attempts to simulate brittle deformation were first made using random spring-like models. Combining local threshold mechanics and long-range elastic interactions, these successfully reproduced the strong localization of rupture in both space and time, the clustering of rupture events along faults and the multifractal properties of strain fields (Cowie et al., 1993, 1995). Building on similar linear-elastic laws and introducing some strain softening at the micro scale, the failure model of Tang (1997) succeeded in simulating the progressive failure leading to the macroscopic non-linear behaviour of brittle rock, thereby processing discontinuum mechanics by a continuum mechanics method. An analogous approach based on local damage evolution was also taken by Amitrano et al. (1999), who combined

– a linear-elastic constitutive relationship for a continuum solid,

– a local Mohr-Coulomb criterion for brittle failure,

– an isotropic progressive damage mechanism for the elastic modulus described by a non-dimensional scalar damage parameter, allowing for the redistribution of the stress from over-critical to sub-critical areas of the material, for the triggering of avalanches of damaging events and the for propagation of faults.

This rheological framework, named Elasto-Brittle (EB), was recently developed in the context of the Arctic ice pack by Girard et al. (2010a, b) to explicitly introduce brittle mechanics concepts in continuum sea ice models. First implementations of this rheology into short (3-days), no-advection, stand-alone simulations of the Arctic, but using realistic wind forcing from reanalyses, showed that the EB model is able to reproduce the strong localization and the anisotropy of damage within sea ice and agrees very well with the deformation fields estimated from the RADARSAT Geophysical Processor System (RGPS) data (Girard et al., 2010b).

In the context of longer-term simulations of ice conditions and coupling to an ocean component, a suitable sea ice model however needs to represent not only the small deformations associated with the fracturing of the pack, but also the permanent deformations occurring once it is fragmented and undamaged ice floes move relative to each

**TCD**

doi:10.5194/tc-2015-200

**A Maxwell–Elasto-Brittle rheology for sea ice modelling**

V. Dansereau et al.

other along open leads, as these much larger deformations set its overall drift patterns and advective processes. This last point is an important and intrinsic limitation of the EB framework, since the linear-elastic constitutive law does not allow solving for the elastic (reversible) and permanent deformations of the simulated material separately.

Hence to estimate the material's velocity, assumptions about the amount of reversible vs. irreversible deformation need to be made in the EB model. The partitioning is bounded by two limit cases. (1) If a loading stress is applied to the damaged material (see Fig. 1b, dashed blue loading path) and all of the resulting deformation is assumed elastic, the material goes back to its initial position if unloaded and its veloc-
ity is zero (red dashed unloading path). This assumption was made in the no-advection simulations of Girard et al. (2010b). (2) Alternatively, if all of the resulting deformation is considered permanent, the material keeps its final position if unloaded (Fig. 1b, purple dashed unloading path) and the velocity is trivially estimated as the ratio of the total deformation and of the time associated with the loading. In the case of sea ice, the
second assumption might be justified by the fact that elastic deformations within an undamaged pack are small compared to the permanent deformations associated with the opening, closing, and shearing along leads. Considering the maximum in-situ values of shear stress of $10^5$ Pa reported by Weiss et al. (2007) and an undamaged elastic modulus between 1.0 and $10.0 \times 10^9$ Pa (Timco and Weeks, 2010), upper bound values
for shear strains in a one meter thick elastic ice pack would be on the order of $10^{-5}$. On daily time scales, these are at the lower bound of RGPS deformation rate estimates (between $10^{-4}$ and $10^0$ day$^{-1}$, for Marsan et al., 2004; Girard et al., 2009), suggesting a dominant contribution of irreversible deformations. This second assumption is taken in the recently developed neXtSIM sea ice model, which is based on the EB rheology
and does represent advective processes over the Arctic (Bouillon and Rampal, 2015). However in this all-permanent deformations limit, internal stresses are immediately dissipated, hence the memory of the stresses associated with elastic deformations is erased whenever the applied loading is removed or reset. Without carrying the history of previous stresses, the model cannot exhibit the intermittency intrinsic to the mechan-

**TCD**

doi:10.5194/tc-2015-200

**A Maxwell–Elasto-Brittle rheology for sea ice modelling**

V. Dansereau et al.

ical behaviour of sea ice, i.e. not directly inhered from the wind forcing (Rampal et al., 2009). In order to estimate adequate drift velocities, a suitable rheological model must therefore have the capacity to distinguish between reversible and irreversible deformations.

The goal of this work is to develop such a model allowing a passage between the small/elastic and large/permanent deformations and with the capability of damage mechanics models to reproduce the observed space and time scaling properties of sea ice deformation. Our approach consists in introducing a viscous relaxation term into the linear-elastic constitutive law of the original EB framework. The new constitutive

relationship takes the form of the Maxwell viscoelastic model. The all-important difference with respect to the Maxwell framework however is that the viscosity associated with the stress dissipation term is not meant to represent the viscoplastic creep of bulk ice (Duval et al., 1983), but instead is an "apparent" viscosity that depends on the local level of damage and concentration of the ice cover. As the elastic modulus, this me-

chanical parameter is coupled to the progressive damage mechanism through a scalar variable $d$ representing the time and space-evolving level of damage of the ice pack. The coupling is designed so that stresses induce elastic strains over undamaged portions of the ice and are dissipated through permanent deformations where the pack is highly fractured.

The use of a viscoelastic rheology and apparent viscosity in the case of sea ice can be supported again by the similarity between the mechanical behaviour of the ice pack and that of the Earth crust and the existence of similar approaches to model lithospheric faulting. Active faults in the Earth crust have been known to deform in two distinct ways: either abruptly, causing earthquakes, or in an transient, aseismic manner

(Scholz, 2002; Gratier et al., 2014; Cakir et al., 2012; Cetin et al., 2014). Similar to sea ice, co-seismic fracturing activates aseismic creep, leading to deformations that can be much larger than that associated with the fracturing itself and to the relaxation of a significant amount of elastic strain (Cakir et al., 2012; Cetin et al., 2014). A further justification of the introduction of such pseudo-viscosity comes from the rheology of

Discussion Paper | Discussion Paper | Discussion Paper | Discussion Paper |

**TCD**

doi:10.5194/tc-2015-200

**A Maxwell–Elasto-Brittle rheology for sea ice modelling**

V. Dansereau et al.

granular media. As sea ice along leads (see Fig. 3), rocks along active faults are highly fragmented. Sheared granular media flow in a viscous manner when inertial effects can be neglected (Jop et al., 2006) with an apparent viscosity diverging as the packing fraction approaches the close-packed limit (Aranson and Tsimring, 2006). This last point will justify the dependence of our apparent viscosity on sea ice concentration.

Viscous-elastic rheological models using apparent viscosities have already been used to model the deformation of rock-like materials. Lyakhovsky et al. (1997) built a viscoelastic damage rheology model with the intent of representing the different stages of geological faulting, from subcritical crack growth to increasing crack concentration and material degradation, macroscopic brittle failure, post failure deformation and healing. However, the evolution of damage in their model was derived from energy conservation principles rather than from a brittle failure criterion and was coupled to the elastic modulus only. Frederiksen and Braun (2001) successfully simulated strain localization during lithospheric extension using an elasto-visco-plastic model together with an ad hoc viscosity. As their work was concerned with the ductile rather than the brittle deformation regime, strain softening in their model did not involve a progressive damage mechanism but instead was achieved by coupling the viscosity to the accumulated strain and the elastic modulus of the material was kept constant. Hamiel et al. (2004) modified the coupled linear elasticity and progressive damage rheological framework of Lyakhovsky et al. (1997) with a non-linear damage-elastic moduli relation and by adding a damage-dependent Maxwell-like viscous term to account for the gradual accumulation of irreversible strain observed in typical rock mechanics experiments. The addition of this term had however a fundamentally different purpose than in the present approach in that it was intended for the representation of the small pre-macroscopic brittle failure deformations, not to bridge between small and large deformations.

To our knowledge, it is therefore the first time a viscoelastic Maxwell constitutive law is coupled to a progressive damage (and healing) mechanism through *both* the elastic modulus and an apparent viscosity with the intent of reproducing the small deformation associated with brittle fracturing and the large, permanent post-fracture deformation of

Discussion Paper | Discussion Paper | Discussion Paper | Discussion Paper |

**TCD**

doi:10.5194/tc-2015-200

**A Maxwell–Elasto-Brittle rheology for sea ice modelling**

V. Dansereau et al.

geomaterials. It is certainly the first time such a rheological model has been adapted for sea ice modelling.

The paper is structured as follow: the Maxwell–EB rheological framework is described in Sect. 2. A dynamical Maxwell–EB sea ice model is presented in Sect. 3 along with its adimensional version and a discussion of the important non-dimensional numbers involved in the model. The numerical scheme employed in the case of small-deformation experiments is presented and idealized model simulations are described in Sect. 4. In Sect. 5, these simulations are analyzed and discussed on the basis of the macroscopic behaviour and convergence properties of the model and of the heterogeneity, anisotropy and intermittency of the simulated deformation. Conclusions are summarized in Sect. 6.

## 2 The Maxwell–EB model

### 2.1 Constitutive relationship

The Maxwell rheology describes the behaviour of a continuum material exhibiting both elastic and viscous properties and combines a Newtonian viscous fluid-like damper and a linear elastic term, typically represented by a spring and dashpot connected in series (see Fig. 1a). Considering the material, typically an incompressible fluid, as being isotropic at the elementary scale for both elastic and viscous properties and assuming plane stress conditions, the Maxwell constitutive relationship reads

$$\frac{1}{G}\frac{D\boldsymbol{\tau}}{Dt} + \frac{1}{\eta}\boldsymbol{\tau} = 2\dot{\varepsilon}(\mathbf{u})$$

with $\boldsymbol{\tau}$, the deviatoric part of the Cauchy stress tensor, $G$ and $\eta$ the (shear) elastic modulus and viscosity of the material associated to the spring and dashpot components respectively, $\dot{\varepsilon}$ the strain rate tensor, defined in terms of the velocity $\boldsymbol{u}$. The objective derivative of the stress tensor is given by

Discussion Paper | Discussion Paper | Discussion Paper | Discussion Paper |

**TCD**

doi:10.5194/tc-2015-200

**A Maxwell–Elasto-Brittle rheology for sea ice modelling**

V. Dansereau et al.

$$\frac{\mathcal{D}\boldsymbol{\tau}}{\mathcal{D}t} = \frac{\partial \boldsymbol{\tau}}{\partial t} + (\boldsymbol{u} \cdot \nabla)\boldsymbol{\tau} + \beta_a(\nabla \boldsymbol{u}, \boldsymbol{\tau})$$

with $\beta_a(\nabla \boldsymbol{u}, \boldsymbol{\tau}) = \boldsymbol{\tau} W(\boldsymbol{u}) - W(\boldsymbol{u})\boldsymbol{\tau} - a(\boldsymbol{\tau} D(\boldsymbol{u}) + D(\boldsymbol{u})\boldsymbol{\tau})$, $D(\boldsymbol{u}) = \frac{\nabla \boldsymbol{u} + \nabla \boldsymbol{u}^T}{2}$ and $W(\boldsymbol{u}) = \frac{\nabla \boldsymbol{u}^T - \nabla \boldsymbol{u}}{2}$ the symmetric and anti-symmetric parts of the velocity gradient and $a = 0, 1$ or $-1$ if using the Jaumann, upper convected or lower convected objective derivative.

When a stress $\boldsymbol{\tau}$ is applied to the Maxwell system, the resulting deformation $\varepsilon_{\text{total}}$ is split between two components: the instantaneous, reversible, deformation of the spring, $\varepsilon_E$, and the permanent deformation of the dashpot, $\varepsilon_\eta$, increasing linearly in time (see Fig. 1a). For a given total deformation applied to the system, the rate of dissipation of the associated stress through the permanent deformation of the dashpot is determined by the ratio, $\frac{\eta}{G}$, of the viscosity of the dashpot and of the (shear) elastic modulus of the spring, $G$. This ratio can be interpreted as a characteristic memory time for elastic deformations: as it decreases, the material looses its capacity to retain the memory of recoverable deformations.

Here we apply the idea of stress dissipation to a compressible, elastic continuum solid and formulate the following constitutive equation by adding a Maxwell-like viscous damper term to the linear elasticity (i.e. Hooke's law) constitutive relationship:

$$\frac{1}{E}\left[\frac{\partial \boldsymbol{\sigma}}{\partial t} + (\boldsymbol{u} \cdot \nabla)\boldsymbol{\sigma} + \beta_a(\nabla \boldsymbol{u}, \boldsymbol{\sigma})\right] + \frac{1}{\eta}\boldsymbol{\sigma} = \mathbf{K} : \dot{\varepsilon}(\boldsymbol{u}) \tag{1}$$

where $\boldsymbol{\sigma}$ is the total Cauchy stress tensor, $E$ is the elastic (or Young) modulus and $\mathbf{K}$ is the (adimensional) elastic stiffness matrix, which in terms of the Poisson ratio $\nu$ writes $\mathbf{K} = \frac{\nu}{(1+\nu)(1-2\nu)}\boldsymbol{I} \otimes \boldsymbol{I} + \frac{2}{2(1+\nu)}\boldsymbol{I}$ with $\boldsymbol{I}$ the second rank identity tensor and $\boldsymbol{I}$ the symmetric part of the fourth rank identity tensor. In this rheological framework, the mechanical parameter $\eta$ is *not* the true dynamic viscosity of the material but rather is an "apparent" viscosity. The related relaxation time, $\lambda = \frac{\eta}{E}$, characterizes the rate at which internal stresses dissipate into permanent deformations.

Discussion Paper | Discussion Paper | Discussion Paper | Discussion Paper | Discussion Paper

**TCD**

doi:10.5194/tc-2015-200

**A Maxwell–Elasto-Brittle rheology for sea ice modelling**

V. Dansereau et al.

## 2.2 Damage criterion

In agreement with in-situ stress measurements (Weiss et al., 2007), and as in the original EB model, the damage criterion in the Maxwell–EB rheology is based on the Mohr-Coulomb (MC) theory of fracture. In terms of the principal stress components $\sigma_1$ and $\sigma_2$, and using the rock mechanics convention that compressive stresses are positive, the MC criterion reads

$$\sigma_1 = q\sigma_2 + \sigma_c \tag{2}$$

(or $\sigma_2 = q\sigma_1 + \sigma_c$, by symmetry of the criterion along the the $\sigma_1 = \sigma_2$ axis – see Fig. 2). The slope of the envelope in the principal stresses plane, $q$, is expressed in terms of the internal friction coefficient $\mu$ as

$$q = \left[ (\mu^2 + 1)^{1/2} + \mu \right]^2 . \tag{3}$$

The intercept $\sigma_c$ of the MC criterion with the $\sigma_1$ axes (see Fig. 2), interpreted as the uniaxial (unconfined) compressive strength, is given by

$$\sigma_c = \frac{2C}{\left[ (\mu^2 + 1)^{1/2} - \mu \right]} . \tag{4}$$

with the cohesion $C$ setting the local resistance of the material to pure shear. Disorder is introduced in the damage criterion through the spatial distribution of $C$. This noise represents the material's natural heterogeneity that causes progressive failure behaviour (e.g. Amitrano et al., 1999; Herrmann and Roux, 1990; Tang, 1997) under homogeneous forcing conditions and is associated with structural defects at the sub-grid scale, thermal cracks in sea ice for instance, serving as stress concentrators (Schulson and Duval, 2009). No correlation length is associated to these heterogeneities, hence their spatial scale corresponds to the spatial resolution of the model, $\Delta x$ (Hutchings et al.,

Discussion Paper | Discussion Paper | Discussion Paper | Discussion Paper

**TCD**

doi:10.5194/tc-2015-200

**A Maxwell–Elasto-Brittle rheology for sea ice modelling**

V. Dansereau et al.

2005; Bouillon and Rampal, 2015), and the value of $C$ over each model element is drawn randomly from a uniform distribution of values spanning estimates from in-situ stress measurements in Arctic sea ice (Weiss et al., 2007). The internal friction coefficient $\mu$ is set to 0.7, a value seemingly scale-independent and consistent with laboratory experiments on Coulombic shear faults in fresh ice (Schulson et al., 2006; Fortt and Schulson, 2007; Weiss and Schulson, 2009) and also common for geomaterials (Byerlee, 1978; Jaeger and Cook, 1979).

For metals and rocks, the MC theory was shown to be defective in the case of tension (Paul, 1961), as the mechanism of tensile failure is intrinsically different to that of compressive failure and, in general, does not involve friction. In the case of $\sigma_1, \sigma_2 < 0$, fracture occurs whenever $\sigma_1$ or $\sigma_2$ reaches a critical value. However, in-situ stress measurements in Arctic sea ice have revealed that pure tensile failure does not significantly modify the Coulombic-like failure envelope of pack ice and that Coulomb branches well describe this envelope even under large tensile stresses, up to at least $\sigma_N \sim 50$ kPa (Weiss et al., 2007). Here, we therefore extend the Mohr-Coulomb criterion to tensile stresses and for practical reasons, set the critical value to the ultimate tensile stress $\sigma_t$, defined as the intersection of the Mohr-Coulomb criterion with the $\sigma_2$ axis (Paul, 1961), as shown on Fig. 2. The tensile strength cutoff therefore takes the form:

$$\sigma_1 < 0; \ \sigma_2 = \sigma_t, \tag{5}$$

where

$$\sigma_t = -\frac{\sigma_c}{q} = -2C \left[ (\mu^2 + 1)^{1/2} + \mu \right]. \tag{6}$$

This gives a ratio of the ultimate tensile stress and uniaxial compressive stress of $\frac{\sigma_t}{\sigma_c} \approx$ 0.27, which might slightly overestimate the tensile strength for sea ice as measured on the field (Weiss et al., 2007) and in the lab (Schulson, 2006) ($\sigma_t \approx 0.2\sigma_c$). However, as such large values of tensile strength are rarely obtained in the Maxwell–EB model simulations, this choice does not significantly affect our results.

**TCD**

doi:10.5194/tc-2015-200

**A Maxwell–Elasto-Brittle rheology for sea ice modelling**

V. Dansereau et al.

No truncation to the MC criterion is used to close the envelope towards biaxial compression (i.e. beyond $\sigma_c$) as instances of large biaxial compressive stresses are seldom encountered in Arctic sea ice (Weiss et al., 2007). Besides, imposing a truncation was shown to have little impact on the simulation results. The damage criterion combining the MC envelope and the tensile strength cutoff is represented in Fig. 2 in the principal stresses plane and has the same shape as deduced by Coon et al. (2007) from measurements in undamaged pack ice.

## 2.3 Progressive damage mechanism and healing

The Maxwell–EB rheology differs from the standard Maxwell rheology in that the mechanical parameters $E$, $\eta$ and $\lambda$ are not constant but all coupled to the spatially and temporally evolving level of damage of the material, which controls its local degradation and re-increase in strength. Consistent with previous damage rheological frameworks, the level of damage is represented by a non-dimensional, scalar parameter $d$ evolving between 1 (undamaged) and 0 ("completely damaged" material). This variable is interpreted as a measure of sub-grid cell defects or crack density (Kemeny and Cook, 1986) and is allowed to evolve through two competing mechanisms: damaging and healing. On the one hand, damaging represents fracturing and the opening of faults, or "leads" in the case of sea ice, occurring when and where the internal stress exceeds the mechanical resistance of the material and which leads to its weakening. Healing on the other hand represents the reconsolidating and strengthening of the damaged material through sintering or, in the case of sea ice, refreezing within open leads. Although this mechanism also contributes to the increase in elastic stiffness ($E \times h$) and effective apparent viscosity ($\eta \times h$) of the ice, healing is distinguished from pure thermodynamic growth or dynamically-driven thickness redistribution (e.g. Rothrock, 1975) in that it applies only where and when the material has been damaged. It therefore allows $d$, $E$ and $\eta$ to re-increase at most to their undamaged value; $d^0 = 1$, $E^0$ and $\eta^0$ respectively. Because the two processes operate simultaneously within the simulated material, an

evolution equation for $d$ needs to include both mechanisms. In the following damaging and healing are first treated separately and then combined in a single equation for $d$.

### 2.3.1 Damaging

Contrary to typical sea ice modelling frameworks, no plastic (i.e. normal) flow rule is prescribed when the damage criterion is reached in the Maxwell–EB model. Instead, when the stress locally exceeds the critical stress, the elastic modulus is allowed to drop, leading to local strain softening (e.g. Amitrano et al., 1999; Cowie et al., 1993; Tang, 1997; Hamiel et al., 2004, and others). Because of the long-range interactions within the elastic medium, local drops in $E$ imply a stress redistribution that can in turn induce damaging of neighbouring elements. By this process, "avalanches" of damaging events can occur and damage can propagate within the material over long distances (Amitrano et al., 1999; Girard et al., 2010a). As the elastic perturbation generated by such events is anisotropic (Eshelby, 1957), this propagation mechanism naturally leads to the emergence of both spatial heterogeneity and anisotropy in the stress and strain fields, i.e. to the formation of linear-like faults (see Sect. 5).

In the Maxwell–EB model, the *change* in level of damage corresponding to a local damage event is determined as a function of the distance of the damaged model element to the yield criterion. Three important assumptions are made when calculating this distance, denoted $d_{\text{crit}}$. The first is that the deformation of each model element is conserved during a damaging event, i.e. at initiation, damage modifies only the local state of stress, not strains. The second is that, for a sufficiently small model time step $\Delta t$, i.e. very small compared to the viscous relaxation time $\lambda$ (see Sect. 3.1), a negligible part of the stress is dissipated into viscous deformation. A third constraint is based on the fact that stresses outside the failure envelope are not physical because brittle failure would occur before the material could support them. Hence we consider that after being damaged, an element has its state of stress lying just on the failure envelope. With these assumptions, the following equality holds for each damaged element:

Discussion Paper | Discussion Paper | Discussion Paper | Discussion Paper | Discussion Paper |

**TCD**

doi:10.5194/tc-2015-200

**A Maxwell–Elasto-Brittle rheology for sea ice modelling**

V. Dansereau et al.

$$\varepsilon' = \varepsilon \longleftrightarrow \frac{\mathbf{K}^{-1}\sigma'}{E \times d_{\text{crit}}} = \frac{\mathbf{K}^{-1}\sigma}{E},$$

where the superscript $'$ denotes the post-damage state of deformation and stress. In terms of the principal stress components, the change in level of damage of a given element is given by

$$d_{\text{crit}} = \frac{\sigma'_1}{\sigma_1} = \frac{\sigma'_2}{\sigma_2}, \tag{7}$$

which implies that as the level of damage varies, all stress components vary in the same proportions. Hence the state of stress $\sigma'$ after each damaging event is given by the intersection of the failure envelope and of the line connecting the pre-damage state of stress $(\sigma_1, \sigma_2)$ with the origin, in the principal stress plane (see Fig. 2). Two cases must be distinguished when calculating $\sigma'$, depending on which of the Mohr-Coulomb or tensile criterion has been exceeded. Combining the two, $d_{\text{crit}}$ is evaluated simultaneously over all mesh elements of the model domain as:

$$d_{\text{crit}} = \min\left[1, \frac{\sigma_t}{\sigma_2}, \frac{\sigma_c}{\sigma_1 - q\sigma_2}\right]. \tag{8}$$

Following progressive damage models, the level of damage of a given element in the Maxwell–EB model at any given time is determined by both its instantaneous distance to the damage criterion $d_{\text{crit}}$, i.e. its current state of stress, and its previous damage level. This implies that the variable $d$ carries the entire history of damage of model elements and, if discretizing time as $t_n = n\Delta t$, $n \geq 0$, translates into the discrete recursive equation

$$d^{n+1} = d_{\text{crit}}^{n+1} d^n, \quad 0 < d^0 \leq 1.$$

**TCD**

doi:10.5194/tc-2015-200

**A Maxwell–Elasto-Brittle rheology for sea ice modelling**

V. Dansereau et al.

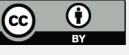

A continuous evolution equation for $d$ can be obtained by considering that the time characterizing the redistribution of stress between model elements is intrinsically tied to the speed of propagation of elastic waves, $c$, in the material, which carry the damage information. Using a Backward explicit scheme of order 1, and setting the model time step to $\Delta t = t_d$ with $t_d = \frac{\Delta x}{c}$, the exact time of propagation of an elastic wave with speed $c$ over a distance $\Delta x$, the following equation arises:

$$\frac{Dd}{Dt} = \frac{d_{\text{crit}} - 1}{t_d} d. \tag{9}$$

### 2.3.2 Healing

By healing, the simulated material is allowed to regain some strength. The characteristic time for this process is designated in the following by $t_h$. It corresponds to the time required for a completely damaged element ($d = 0$) to recover its initial stiffness ($d = 1$), which in a dynamic-thermodynamic sea ice model would depend on the local difference between the temperature of the air near the surface of the ice and the freezing point of seawater below. Healing schemes of varying level of complexity could be used in the Maxwell–EB model. One possibility is the one employed in the EB sea ice model of Girard et al. (2010a), which follows parameterizations of the vertical growth of sea ice (Maykut, 1986). An underlying assumption is that the rate of healing is inversely proportional to the level of damaging of the ice. However as there is no physical evidence for this assumption, in the following, uncoupled, implementation of the Maxwell–EB model we use an even simpler parameterization that implies a constant healing rate, $\frac{1}{t_h}$:

$$\frac{Dd}{Dt} = \frac{1}{t_h}, \ 0 \le d \le 1. \tag{10}$$

Combining both the damaging and healing mechanisms (Eqs. 8–10), the complete evolution equation for $d$ is

Discussion Paper | Discussion Paper | Discussion Paper | Discussion Paper |

**TCD**

doi:10.5194/tc-2015-200

**A Maxwell–Elasto-Brittle rheology for sea ice modelling**

V. Dansereau et al.

$$\frac{\partial d}{\partial t} + (\boldsymbol{u} \cdot \nabla)d = \left( \min\left[ 1, \frac{\sigma_t}{\sigma_2}, \frac{\sigma_c}{\sigma_1 - q\sigma_2} \right] - 1 \right) \frac{1}{t_d}d + \frac{1}{t_h}, \quad 0 < d \leq 1. \tag{11}$$

Although the two processes apply simultaneously on the level of damage in the model, they are inherently distinct. On the one hand, damaging is a discrete threshold mechanism applying only where and when the state of stress becomes overcritical. As mentioned in Sects. 2.2 and 2.3.1, the characteristic time for this process, $t_d$, is tied to the speed of propagation of (shear) elastic waves and to the model's spatial resolution. In the case of an heterogeneous ice pack, an average value for $c$ is on the order of $500\,\mathrm{m\,s^{-1}}$ (Marsan et al., 2011). For spatial resolutions between that of current global climate and high resolution regional sea ice models ($\Delta x = 1$ to $100\,\mathrm{km}$), the characteristic time for damaging, $t_d$ therefore varies between $O(1)$ and $O(10^2)$ s. Healing on the other hand is a continuous process acting on all model elements, independently of the local distance to the damage criteria. Studies on the refreezing within leads in sea ice showed that the time for $1\,\mathrm{m}$ of ice to grow within an opening of $10\,\mathrm{cm}$ under atmospheric temperatures of $T_a = -15\,°\mathrm{C}$ is of $O(100)$ hours or $O(10^5)$ seconds (Petrich et al., 2007). The orders of magnitude of difference between $t_h$ and $t_d$ therefore imply that the two processes are intrinsically decoupled in the case of the ice pack.

### 2.3.3 Coupling $d$ with $E$ and $\eta$

The coupling between the Maxwell–EB constitutive relationship and the progressive damage mechanism constitutes one of the main features of this new modelling framework. It is defined such that:

- Deformations within an undamaged medium are small and reversible, i.e. strictly elastic. Hence undamaged portions of the simulated material have a maximum elastic modulus $E^0$ and a very large apparent viscosity $\eta^0$. In this case, the viscous term in Eq. (3) is negligible and a linear-elastic constitutive relationship is recovered (Fig. 3, right panel).

Discussion Paper | Discussion Paper | Discussion Paper | Discussion Paper |

**TCD**

doi:10.5194/tc-2015-200

**A Maxwell–Elasto-Brittle rheology for sea ice modelling**

V. Dansereau et al.

– Deformations can accumulate over highly damaged areas of the material to become arbitrarily large. These deformations are permanent and dissipate most of the the stress applied to the material within a short relaxation time. Hence the elastic modulus, viscosity and relaxation time drop locally over damaged areas. In the limit of a completely damaged material, elastic interactions are hindered and deformations are strictly irreversible (Fig. 3, left panel). In this case, $\lambda \longrightarrow t_{\mathrm{d}}$ and a soft elastic-plastic behaviour is recovered in which the memory of the elastic stresses is totally lost (narrow-dashed blue line on Fig. 1).

– As damaged areas are allowed to heal, $E$, $\eta$ and $\lambda$ all re-increase, up to their initial undamaged values.

Different functions could be used to express the dependence of $E$, $\eta$ and $\lambda$ on $d$ that meet these criteria. In the absence of physical evidences for a higher level of complexity, and consistent with the relationship between the elastic modulus and crack density used in damage models of rocks (Agnon and Lyakhovsky, 1995; Amitrano et al., 1999; Schapery, 1999), we use the simplest parameterization and set

$$E(t) = E^0 d(t)$$
$$\eta(t) = \eta^0 d(t)^{\alpha},$$

such that

$$\lambda(t) = \frac{\eta^0}{E^0} d(t)^{\alpha-1}$$

with $\alpha$ a constant greater than 1 introduced to fulfil the constraint that the relaxation time for the stress also decreases with increasing damage and re-increases with healing, as the material respectively looses and recovers the memory of reversible deformations. Using this formulation, both $\eta$ and $E$ are entirely defined by their initial value, a constant, and by the level of damage variable $d$. However, the constitutive equation

**TCD**

doi:10.5194/tc-2015-200

**A Maxwell–Elasto-Brittle rheology for sea ice modelling**

V. Dansereau et al.

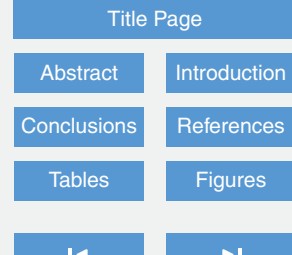

Title Page

| Abstract | Introduction |
| Conclusions | References |
| Tables | Figures |

| ◄◄ | ►► |
| ◄ | ► |
| Back | Close |

Full Screen / Esc

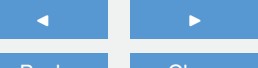

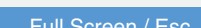

Interactive Discussion

Discussion Paper | Discussion Paper | Discussion Paper | Discussion Paper | Discussion Paper |

becomes undefined in the limit of $d \rightarrow 0$. This problem can be handled by imposing a fixed minimum value $d_{\min} > 0$ for the level of damage. Alternatively, a cutoff $\eta_{\min} \ll \eta^0$ on the value of the apparent viscosity can be introduced and the expression for $\eta(d)$ modified as

$$\eta = (\eta^0 - \eta_{\min})d^\alpha + \eta_{\min} = \begin{cases} \eta^0 & \text{for } d = 1, \\ \eta_{\min} & \text{for } d = 0. \end{cases} \tag{12}$$

Substituting for $\eta$ in the expression for the relaxation time, the elastic modulus then becomes

$$E = \frac{\eta^0 - \eta_{\min}}{\eta^0}E^0 d + \frac{\eta_{\min}}{\eta^0}\frac{1}{d^{\alpha-1}}E^0, \tag{13}$$

such that $E \approx E^0$ for $d = 1$, $E$ decreases with $d$ until a minimum at $d(E = E_{\min}) = \left[\frac{\eta_{\min}}{\eta^0 - \eta_{\min}}(\alpha - 1)\right]^{\frac{1}{\alpha}}$ and $E \rightarrow \infty$ for $d \rightarrow 0$. Using such a cutoff on $\eta$, the elastic term in the Maxwell–EB constitutive equation therefore vanishes in the limit of a "totally" damaged material and the rate of viscous dissipation is then set by the minimum viscosity $\eta_{\min}$. It is important to note that this limit has no physical significance in the context of a progressive damage model for a continuum solid and is rather introduced to insure mathematical consistency while retaining a continuous function for the level of damage. In the following implementation of the model, we take this approach instead of imposing a minimum value of $d$, but it had really no impact on our results since in the simulations presented here $d > d(E = E_{\min})$ at all times.

## 3 A Maxwell–EB sea ice model

In this section, the Maxwell–EB rheology is implemented in the context of sea ice modelling. As in regional and global sea ice models, the ice cover is considered as

Discussion Paper | Discussion Paper | Discussion Paper | Discussion Paper |

**TCD**

doi:10.5194/tc-2015-200

**A Maxwell–Elasto-Brittle rheology for sea ice modelling**

V. Dansereau et al.

a 2-dimensional plate due to its very large aspect ratio and a constant healing rate is assumed. In this case, the complete dynamical model is given by the following system of equations:

1. The momentum equation:

$$\rho h \left[ \frac{\partial \boldsymbol{u}}{\partial t} + (\boldsymbol{u} \cdot \nabla)\boldsymbol{u} \right] = \boldsymbol{F}_{\text{ext}} + \boldsymbol{\nabla} \cdot (\sigma h), \tag{14}$$

with $\boldsymbol{u}$ the velocity, $h$ the thickness and $\rho$ the density of sea ice. $\boldsymbol{F}_{\text{ext}}$ assimilates all external forces on the sea ice cover, which in regional and global sea ice models are typically the air and ocean drags and the forces associated with the Coriolis acceleration and gradients in sea surface height. We assume the internal stress to be homogeneously distributed within the depth $h$ and following Bouillon and Rampal (2015) and Sulsky et al. (2007), we write the momentum equation in terms of the internal stress rather than the vertically integrated stress tensor commonly used in the sea ice modelling community.

2. Conservation equations for the ice concentration $A$ and ice thickness $h$:

$$\frac{\partial h}{\partial t} + \nabla \cdot (h\boldsymbol{u}) = S_h, \tag{15}$$

$$\frac{\partial A}{\partial t} + \nabla \cdot (A\boldsymbol{u}) = S_A, \tag{16}$$

where $S_h$ and $S_A$ represents thermodynamic source and diffusion terms and elastic compressibility effects are assumed negligible relative to dynamic variations of the ice volume in the conservation of the mass of the sea ice cover.

3. The constitutive relationship (Eq. 1) with

$$E = f_1(E^0, \eta^0, \eta_{\min}, d)e^{-c(1-A)}, \tag{17}$$

$$\eta = f_2(\eta^0, \eta_{\min}, d)e^{-c(1-A)}, \tag{18}$$

where $f_1$ and $f_2$ represent the functional dependance on the level of damage of the ice $d$, given by Eqs. (13) and (12) respectively. The exponential function of the ice concentration allows the internal stress term to be maximal when $A = 100\%$ and to decrease rapidly when leads open and $A$ drops. It is of the same form as that used for the pressure term ($P$, or ice strength in compression) in the VP rheology of Hibler (1979). Here the non-dimensional parameter $c*$ characterizing this dependence on the ice concentration has the same (constant) value for both mechanical parameters, but could be set different in a refined parameterization.

4. The equation for the evolution of damage Eq. (11) with the damage criterion defined by Eqs. (2) and (5) and $q$, $\sigma_c$ and $\sigma_t$ given by Eqs. (3), (4) and (6) in terms of the cohesion variable $C$ and of the constant internal friction coefficient $\mu$.

In the case of "quenched disorder" (i.e. when the field of $C$ is set at the beginning of a model simulation), an additional equation arises that handles the advection of the field of cohesion with the simulated velocity field. Table 1 lists all model variables and parameters.

## 3.1 Characteristic numbers and times

Neglecting all thermodynamic effects and variations in ice thickness and concentration (considering $h = 1$ and $A = 100\%$) as well as external forcings and adimensionalizing with respect to the ice velocity $U$, the horizontal extent of the model domain $L$, the thickness of the ice $H$ and the undamaged elastic modulus $E^0$, the dynamical system of equations read

$$Ca^0 \frac{D\tilde{\boldsymbol{u}}}{D\tilde{t}} = \tilde{\nabla} \cdot \tilde{\sigma} \tag{19}$$

$$We^0 \tilde{d}^{\alpha-1} \frac{\mathcal{D}\sigma}{\mathcal{D}t} + \tilde{\sigma} = We^0 \tilde{d}^{\alpha'} \mathbf{K}(\nu) : \tilde{\tilde{\varepsilon}}(\tilde{\boldsymbol{u}}) \tag{20}$$

Discussion Paper | Discussion Paper | Discussion Paper | Discussion Paper |

**TCD**

doi:10.5194/tc-2015-200

**A Maxwell–Elasto-Brittle rheology for sea ice modelling**

V. Dansereau et al.

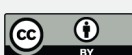

$$\frac{D\tilde{d}}{D\tilde{t}} = \left(\min\left[1, \Sigma_t \frac{1}{\tilde{\sigma}_2}, \Sigma_c \frac{1}{\tilde{\sigma}_1 - q\tilde{\sigma}_2}\right] - 1\right)\frac{1}{T_d}\tilde{d} + \frac{1}{T_h}, \quad 0 < \tilde{d} \le 1. \tag{21}$$

where $\tilde{d}^{\alpha'} = \left[\left(1 - \frac{\eta_{min}}{\eta^0}\right)\tilde{d}^{\alpha} + \frac{\eta_{min}}{\eta^0}\right]$ and the superscript " $\tilde{\ }$ " is used for all non-dimensional variables and operators.

In this form, the model involves 8 characteristic numbers and time scales, some of which are constant and some, evolving with the local level of damage $d$: $Ca^0$, the (undamaged) Cauchy number, $We = We^0\tilde{d}^{\alpha-1}$, the Weissenberg number, with $We^0$ its undamaged value, $\nu$ Poisson's ratio, $\Sigma_t$ the dimensionless critical tensile stress, $\Sigma_c$ the dimensionless critical stress with respect to the Mohr–Coulomb criterion, $T_d$ the characteristic time for damaging, $T_h$ the characteristic time for healing and $\alpha$ the damage constant. In order for the Maxwell–EB model to represent the intended physics, the value of these parameters must evolve within a certain range of values. In the following we elaborate on the absolute and relative values of those numbers which are the most critical in the context of sea ice modelling.

### 3.1.1 The characteristic time for damaging, $T_d$

As mentioned in the previous section, the (adimensional) characteristic time for the propagation of damage, $T_d = \frac{t_d}{T}$ with $T = \frac{L}{U}$, is determined by the speed of propagation of elastic waves within the simulated material and is strongly tied to the mean spatial resolution of the model, as $t_d$ should be of $O(\frac{\Delta x}{c})$. In turn, this time places a strong constraint on the Maxwell–EB model time step. Setting $\Delta t < \frac{\Delta x}{c}$ is indeed unphysical, as the time associated to one model iteration would then be too short for the stress to be redistributed from one overcritical element to its direct neighbour. For the model to resolve the propagation of damage, the time step must therefore be $\ge t_d$.

No strict upper bound to $\Delta t$ is imposed by the damage mechanism. One the one hand, choosing $\Delta t > t_d$ could be interesting in terms of reducing computational costs. Physically, it implies that damage is allowed to propagate beyond the first neighbour

Discussion Paper | Discussion Paper | Discussion Paper | Discussion Paper |

**TCD**

doi:10.5194/tc-2015-200

**A Maxwell–Elasto-Brittle rheology for sea ice modelling**

V. Dansereau et al.

**[TCD]**

doi:10.5194/tc-2015-200

**A Maxwell–Elasto-Brittle rheology for sea ice modelling**

V. Dansereau et al.

barrier and over larger distances within one model time step. On the other hand, increasing $\Delta t$ with respect to $t_d$ also implies (1) a decrease in the resolution of damaging, as the model might miss important intermediate damage events that trigger additional interactions between neighbouring elements and (2) larger local drops in the level of damage, inducing large stress perturbations and, potentially, numerical instabilities in the model. Sensitivity analyses on the propagation of the damage should therefore be performed when choosing $\Delta t > t_d$. The temporal resolution that is optimal in terms of capturing all elastic interactions within the simulated material is therefore $\Delta t = t_d$. In the model experiments presented in the following, this is the choice we make.

### 3.1.2 The characteristic time for healing, $T_h$

In order for healing not to offset damaging in the rate of change of $d$, the (adimensional) time for healing, $T_h = \frac{t_h}{T}$, must be much larger than the (adimensional) time for damage propagation. This separation of scales ensures that elements cannot recover by healing more strength than they have lost by damaging within one time step, as excess healing would effectively entail a net growth of the material, a process that is not intended by this parameterization and should instead be accounted for by thermodynamic, balance calculations. Considering the estimates of the speed of elastic waves and of the healing rate of leads aforementioned (Sect. 2.3), pack ice naturally meets this condition.

### 3.1.3 The Weissenberg number, *We*

The Weissenberg number, *We*, defined as the dimensionless product of the viscous relaxation time for the stress and of time $T = \frac{L}{U}$ characterizing the deformation process:

$$We = \frac{\eta}{E}\frac{U}{L} = \frac{\lambda}{T},\tag{22}$$

sets the viscous vs. elastic character of the flow of a viscoelastic material. In the original Maxwell model, $We = 0$ represents the limit of zero elastic stresses, while a very large

*We* characterizes a strictly elastic solid. In the Maxwell–EB model, the Weissenberg number evolves according to the level of damage as $We = We^0 d^{\alpha-1}$ with $We^0$, its maximum value.

As viscous dissipation should be insignificant over undamaged and strictly elastic areas of the material, $We^0$ should be chosen very large, representing the limit of $\frac{1}{\eta^0} \longrightarrow 0$. In this case the viscous term in the constitutive relationship Eq. (1) effectively vanishes and a linear elastic rheology is recovered. In practice, the value of $We^0$ is however limited, first, by the machine precision and second, due to a numerical scheme failure known in the field of viscoelastic flow computations as the high Weissenberg number problem (Keunings, 1986; Fattal and Kupferman, 2004, 2005; Saramito, 2014). For large values of $We$, numerical instabilities arise in Maxwell-type models due the presence of deformation source terms ($\beta_a$) in the transport equation for the stress tensor euqation. With $We^0$ (or equivalently, $\lambda^0$) too low, simulations can run for a time $t \sim \lambda^0$ and unphysical viscous dissipation can occur over undamaged parts of the simulated material. To get round this problem, the viscous term in the Maxwell constitutive relationship can be multiplied by a Heaviside function $d^*$ that effectively sets $\frac{1}{\eta}$ to the limit value of 0 when and where $d \geq d_c$, with $d_c$ a chosen threshold value (e.g. $d_c = 1$ when using a constant heal rate parameterization) and leaves the constitutive equation unchanged ($d^* = 1$) otherwise. In small-deformation experiments, i.e. run for a time $t \ll \lambda^0$, viscous dissipation over undamaged parts of the material is not significant and the inclusion of such a function, unnecessary.

Conversely, where damage becomes important, the viscous relaxation time $\lambda$ should decrease significantly below the characteristic time for healing to allow for internal stresses to "have time" to dissipate and deformations to become large.

### 3.1.4 The Cauchy number, *Ca*

The dimensionless number that arises when adimensionalizing stresses in the momentum equation with respect to the elastic modulus is the Cauchy number, defined as the

Discussion Paper | Discussion Paper | Discussion Paper | Discussion Paper |

**TCD**

doi:10.5194/tc-2015-200

**A Maxwell–Elasto-Brittle rheology for sea ice modelling**

V. Dansereau et al.

ratio of inertial to elastic forces ($Ca = \frac{\rho U^2}{E}$). If inertial forces are comparable to elastic forces and $Ca \sim 1$, the effect of the propagation of viscoelastic waves in the material cannot be neglected. Yet, setting $\Delta t \geq t_{\mathrm{d}}$, that is $\Delta t$ at least equal to the period of shear elastic waves, implies that the model does not resolve these waves, but only their con-
sequence of transmitting the damage information within the material. Hence the wave signal cannot be properly filtered out of the model's solution. In order for the wave contribution not to have a significant effect on the simulated deformation and stress fields, $Ca$ must therefore be $\ll 1$. Dimensional analysis indicates that over an undamaged ice pack with velocity ranging between 0.001 and $1\,\mathrm{m\,s^{-1}}$, $Ca^0$ is in the range $[10^{-12}$–$10^{-6}]$.
Hence inertial effects can be safely neglected. For simulated ice velocities $U < 1\,\mathrm{m\,s^{-1}}$, and $\alpha > 2$, inertial effects in the Maxwell–EB model remain negligible when damage becomes important.

### 3.1.5 The damage parameter, $\alpha$

The damage parameter $\alpha$ controls the rate at which the apparent viscosity decreases and the material looses its elastic properties with damaging. As mentioned in previous sections, it should be set greater than 1 in order for the viscous relaxation time to decrease with damaging. The requirements that (1) the viscous relaxation time drops well below the time for healing over highly damaged areas and (2) inertial effects remain negligible for high deformation rates (i.e. large velocities) can also place a constraint on the minimum value of $\alpha$. Conversely, for large values of $\alpha$, the relaxation time $\lambda$ becomes very small whatever the damage level (see Sect. 2.3.3). This means that elastic deformations are almost immediately dissipated after damaging, that is, the model becomes purely elasto-plastic. For the experiments presented here, we find that $\alpha = 4$ allows representing both the brittle behaviour and the relaxation of the internal stress within a material with mechanical parameters in the range of the values suitable for sea ice. For $\alpha$ larger than about 7, memory effects become insignificant and the experiment

**TCD**

doi:10.5194/tc-2015-200

**A Maxwell–Elasto-Brittle rheology for sea ice modelling**

V. Dansereau et al.

Discussion Paper | Discussion Paper | Discussion Paper | Discussion Paper |

**TCD**

doi:10.5194/tc-2015-200

**A Maxwell–Elasto-Brittle rheology for sea ice modelling**

V. Dansereau et al.

instead exhibits a stick-slip behaviour with a well-defined characteristic frequency (not shown).

## 4   Numerical scheme and experiments

The objective time derivative for the Cauchy stress $\sigma$ in the Maxwell–EB constitutive relationship Eq. (1) is composed of an inertial term, an advection term and of a sum of rotation and deformation ($\beta_a$) terms, each of which implies a different level of numerical complexity. In developing the model, our approach is to introduce each of these terms separately in order to evaluate their respective contribution to the simulated mechanical behaviour. On the one hand, introducing the inertial term while neglecting the advection and $\beta_a$ terms allows retaining a Lagrangian scheme, similar to the original EB model (Girard et al., 2010b). Without any remeshing of the domain, the model is then suitable for short-term, small-deformation simulations only. On the other hand, when permanent deformations accumulate over a long time, the advection term is no longer negligible and $\beta_a$ terms become potentially important.

In the following, we present small-deformation numerical experiments that allow analyzing the mechanical behaviour of the Maxwell–EB model in terms of the statistical and scaling properties of the simulated damage and deformation fields. Performed with a highly idealized configuration for the domain geometry, the applied loading and boundary conditions, these will demonstrate that the main characteristics of sea ice deformation (spatial heterogeneity, anisotropy, intermittency) naturally emerge from the underlying physics and do not need to be implemented in an ad-hoc manner.

The simulations represent the uniaxial compression of a (2-dimensional) rectangular ice plate with dimensions $\frac{L}{2} \times L$ (see Fig. 4a). Compression is applied by prescribing a constant velocity $U$ on the upper short edge of the plate with the opposite edge maintained fixed in the direction of the forcing. No confinement is applied on the lateral sides. The velocity $U$ is set small enough to ensure a low driving rate (i.e. slow compared to time scale of damage propagation, Cowie et al., 1993).

In the present implementation, the model is not yet coupled to a thermodynamic component, hence $S_A = S_h = 0$. As advection is neglected and simulations are run for a short enough time such that the macroscopic and local deformations within the ice cover remain small ($\sim 1\%$ of the area of model elements), dynamics-induced variations (through convergence-divergence) of the ice thickness and concentration are not accounted for and hence the mechanical parameters $E$, $\eta$ and $C$ are not yet coupled to $h$ or $A$. Conservation of mass is therefore not imposed in these small-deformation simulations, equivalent to assuming a uniform, constant thickness (1 m) and ice concentration (100 %). In this case, the system of equations reduces to Eqs. (19) to (21) with $\frac{D\tilde{u}}{D\tilde{t}} = 0$ and the 6 unknowns $\tilde{u}$ (2 components), $\tilde{\sigma}$ (3 components) and $\tilde{d}$. The model is made adimensional with respect to the length of the rectangular plate, $L$, the prescribed velocity $U$ on the top boundary, the undamaged elastic modulus $E_0$.

In all simulations, the time step is set equal to the characteristic time for damage propagation. A semi-implicit scheme is used that linearizes the system, in which the momentum and constitutive equations are first solved simultaneously using a backward Euler scheme of $O(1)$ and the value of $\tilde{d}$ at the previous model time step. The level of damage is updated in a second time using the estimated $\tilde{u}$ and $\tilde{\sigma}$ and an explicit scheme of $O(1)$. A fixed-point algorithm iterates between these two steps until the residual of the linearized constitutive equation drops below a chosen tolerance, ensuring the convergence of the solution. Finite elements and variational methods are used to solve the time-discretized problem on a Lagrangian grid within the C++ environment RHEOLEF (Saramito, 2013: http://cel.archives-ouvertes.fr/cel-00573970). An unstructured mesh with triangular elements is used and the average spatial resolution is set by choosing the number $N$ of elements along the short side of the domain.

All simulations are started from an initially undamaged ice cover with uniform elastic modulus and viscosity. Undamaged mechanical parameter values are chosen so that to be representative of sea ice on regional to global scales ($c = 500\,\mathrm{m\,s^{-1}}$ and $v = 0.3$). The undamaged elastic modulus is given by the relation $E^0 = 2c^2(1 + v)\rho$ and the undamaged viscosity $\eta^0$ is set such that the initial relaxation time $\lambda^0$ is as large

Discussion Paper | Discussion Paper | Discussion Paper | Discussion Paper |

**TCD**

doi:10.5194/tc-2015-200

**A Maxwell–Elasto-Brittle rheology for sea ice modelling**

V. Dansereau et al.

**TCD**

doi:10.5194/tc-2015-200

as possible while the maximum Weissenberg number, $We^0$ is small ($< 1$). All model variables and parameters are listed in Table 1. Parameter values are not varied in any of the experiments presented here as a sensitivity study is kept for a separate paper.

## 5  Results

In this section we analyze the mechanical behaviour of the Maxwell–EB model. In particular, we evaluate its capacity to reproduce the main characteristics of sea ice deformation, which are its spatial heterogeneity, intermittency and anisotropy, following the methodology developed in previous observational studies of the deformation and drift of the Arctic ice pack.

One signature of the strong heterogeneity of sea ice deformation is the emergence of a spatial scaling in the deformation fields over a wide range of scales. Using a coarse-graining procedure, Marsan et al. (2004) performed a scaling analysis of the deformation of sea ice over the Arctic using the 3 days, 10 km × 10 km gridded RGPS deformation product. Doing so, they obtained a power-law relationship between the total deformation rate $< \dot{\varepsilon}_{tot} >_l$ invariant and the corresponding averaging scale $l$ of the form

$$< \dot{\varepsilon}_{tot} >_l \sim l^{-\beta} \tag{23}$$

with a constant exponent $\beta > 0$, indicating correlations in the deformation fields over 2 orders of magnitude in $l$ and an increase in the mean strain rate with decreasing scale of observation, in agreement with a strong spatial localization of the deformation.

This coarse-graining calculation was later extended to ice buoy data (e.g. Rampal et al., 2008; Hutchings et al., 2011) which, with a higher temporal resolution than the RGPS data, allowed performing scaling analyses of Arctic sea ice deformation in the temporal dimension as well. Using the dispersion rate of buoys as a proxy for the strain rate, Rampal et al. (2008) obtained a power-law relationship between the total deformation rate $< \dot{\varepsilon}_{tot} >_t$ computed at a chosen space scale and the time scale of observation $t$

Discussion Paper | Discussion Paper | Discussion Paper | Discussion Paper |

$$< \dot{\varepsilon}_{\text{tot}} >_t \sim t^{-\gamma} \tag{24}$$

with a constant exponent $\gamma > 0$ over 2 orders of magnitudes in $t$ (3 h to 3 months), indicating an increase of strain rates with decreasing temporal scale consistent with an intermittent deformation process. Recently, these temporal and spatial scaling properties have been used as benchmarks to validate (or invalidate) sea ice models (e.g. Girard et al., 2009, 2010a; Bouillon and Rampal, 2015).

An additional and all-important characteristic of the deformation of sea ice that is not captured by these scaling analyses is its strong anisotropy. This property has been made evident since the availability of satellite imagery-derived ice motion products (e.g. Stern et al., 1995), which showed that high strain rates concentrate along oriented, linear-like faults, or leads, often termed "linear kinematic features" (Kwok, 2001).

## 5.1 Spatial resolution, convergence and dependence on the initial conditions

In a first time, we analyze the overall, macroscopic behaviour of the Maxwell–EB model, its convergence properties and the dependance of the solution on the prescribed initial conditions. To do so, a set of four uniaxial compression simulations is run using different spatial resolutions, with $N = 10$, 20, 40 and 80. The values of the initial, undamaged mechanical parameters are identical between the simulations as well as the field of cohesion, which is defined at the lowest resolution ($N = 10$) and interpolated onto the higher resolution mesh grids.

Figure 5 shows the (adimensional) macroscopic stress, $\sigma_{\text{m}}$ (normal stress integrated on the upper boundary of the domain), as a function of the adimensional macroscopic strain, $\varepsilon_{\text{m}}$, set by the prescribed displacement of the upper boundary, for these four simulations. The dotted line represents the damage rate (the number of damaged elements per model time step times their distance to the damage criterion, $1 - d_{\text{crit}}$) for the simulation with $N = 40$. Inspection of the initial loading and damaging sequence suggests that the mechanical behaviour is similar to that obtained with other elasto-brittle

Discussion Paper | Discussion Paper | Discussion Paper | Discussion Paper |

**TCD**

doi:10.5194/tc-2015-200

**A Maxwell–Elasto-Brittle rheology for sea ice modelling**

V. Dansereau et al.

models (e.g. Tang, 1997; Amitrano et al., 1999; Girard et al., 2010a). The Maxwell–EB model simulates

1. A strictly linear-elastic behaviour at the initial stage of the experiment, as the material is initially undamaged.

2. A deviation from the linear-elastic behaviour after the onset of damage (marked by the red dot 1), indicative of macroscopic strain softening, with damage distributed homogeneously throughout the material (see Fig. 5b1).

3. The formation of clusters of damaged elements, non-interacting at first, then joining along linear features. This stage is marked by a rapid increase in the number of damaged elements.

4. A sharp stress drop associated with the macroscopic failure of the sample and propagation of a main fault spanning the entire domain (see Fig. 5b2).

In the Maxwell–EB model, this last stage is characterized by a drop in the Weissenberg number (i.e. in $\lambda$) localized along the main fault (not shown), where strain rates are orders of magnitude higher than over undamaged parts of the material. Then, as damaged areas heal, stress builds up again within the material. At all spatial resolutions, the model simulates cycles of slow stress build ups (healing phase) and rapid stress relaxations (damaging phase).

Because the simulations use the same spatial distribution of the damage criteria (i.e. of $C$) the location of the first damage events is the same at all resolutions, as shown by the maps of instantaneous level of damage $d$ near the onset of damaging (Fig. 5b1). However, soon after these first failure events, model solutions do not converge (Fig. 5b2–4) and fractures form with a shape and orientation differing between simulations. This divergence between the post-damage solutions illustrates an all-important and intrinsic characteristic of the Maxwell–EB framework arising from the fact that there is no physical scale associated with the localization of damage in the model. Through elastic interactions, damage and deformation tend to localize at the

**TCD**

doi:10.5194/tc-2015-200

**A Maxwell–Elasto-Brittle rheology for sea ice modelling**

V. Dansereau et al.

Discussion Paper | Discussion Paper | Discussion Paper | Discussion Paper

finest scale (the mesh element), resulting in a different redistribution of the stress between neighbouring elements at different spatial resolutions and hence a non-identical propagation of the damage. Put another way, the divergence of the solutions indicates that while the disorder in $C$ sets the location of the first damage events, the heterogeneities introduced in the stress field by these events prevail in setting the location and timing of subsequent events. This result is consistent with previous elasto-brittle model simulations which have shown that the number of active faults as well as the degree of localization of the deformation over long time scales do not depend systematically on the disorder initially introduced in the model (Cowie et al., 1993) and that once formed, faults produce their own stress field which dominates further fracture growth (Tang, 1997).

Another important property of the deformation made evident by this set of experiments is its strong anisotropy. The fields of $d$ and of the total deformation ($\dot{\varepsilon}_{tot}$) represented on Fig. 5 indeed show that at all spatial resolutions, the simulated damage and deformation are both highly localized and oriented along linear features. This is an important result, as no anisotropy is introduced at the local scale on either the elastic or viscous properties, or in the damage parameterization. This property arises naturally due to elastic interactions within the material and without the need to prescribe fault orientations. It was reproduced by the original EB model (Amitrano et al., 1999; Girard et al., 2010a, b) and is not lost when including a viscous dissipation term for the stress in the Maxwell–EB constitutive relationship.

## 5.2   Heterogeneity

As shown in the previous section, when simulations are started from an undamaged state, the simulated mechanical behaviour of the material is intrinsically different between the first and subsequent loading and damaging cycles. The path to the first rupture in "irreversible damage" (i.e. models without healing) elasto-brittle models has already been investigated in depth (e.g. Girard et al., 2010a; Tang, 1997; Amitrano

**TCD**

doi:10.5194/tc-2015-200

**A Maxwell–Elasto-Brittle rheology for sea ice modelling**

V. Dansereau et al.

et al., 1999). Hence we focus our analysis of the spatial dependence of the Maxwell–EB model strain rate fields on the post macro-rupture behaviour.

To quantify the heterogeneity of the simulated deformation, we follow Marsan et al. (2004) and estimate deformation rates over two orders of magnitude in space scales using a coarse-graining procedure. The calculation is described in details by Girard et al. (2010a). For this analysis we use the outputs of strain rate fields from simulations with $N = 100$, averaged over a time interval corresponding to the time of propagation of an elastic shear wave with speed $c$ through the width of the domain ($\frac{L}{2} \frac{1}{T \times c} = N$ time steps).

The dependence of the deformation rates on the spatial scale of observation is investigated at different stages of the healing-damaging cycle. Figure 6 (a and b) shows the total deformation rate $<\dot{\varepsilon}_{tot}>_l$ as a function of the space scale $l$ at 5 equally-spaced steps along the path towards a given macroscopic failure event, that is, between the minimum in macroscopic stress that follows the propagation of a fault and the maximum that precedes the next macro-rupture, as indicated in Fig. 6a. Deformation rates are normalized by $<\dot{\varepsilon}_{tot}>$ at the smallest averaging scale ($L/N$). At the first stage, just following the rupture (red curve), the total deformation rate shows a clear power law decrease with increasing spatial scale of the form of Eq. (23) over nearly two orders of magnitude of $l$, consistent with a strong localization of the deformation. At the subsequent stages (yellow and green curves), damaged elements progressively recover their mechanical strength by healing. Deformation rates decrease along the main fault and re-increases over undamaged areas, hence deformation homogenizes over the domain and the rate of decrease of $<\dot{\varepsilon}_{tot}>_l$ with $l$ is reduced. Then, as healing allows stress to build up within the material, damaging resumes and clusters in space and the exponent $\beta$ re-increases towards its post macro-rupture value (blue and purple curves).

Repeating the procedure for subsequent healing and damaging cycles and for multiple realizations of the experiment initialized with different cohesion fields showed a similar evolution of the rate of decrease of $<\dot{\varepsilon}_{tot}>_l$ with $l$ between macro-ruptures events,

Discussion Paper | Discussion Paper | Discussion Paper | Discussion Paper | Discussion Paper |

**TCD**

doi:10.5194/tc-2015-200

**A Maxwell–Elasto-Brittle rheology for sea ice modelling**

V. Dansereau et al.

with values of $\beta$ in the vicinity of the rupture consistent with previous EB model analyses (e.g. Girard et al. (2010a), $\beta = 0.15 \pm 0.02$). However, an important difference between the present results and that of Girard et al. (2010a) is the absence of a clear cross-over scale for which $< \dot{\varepsilon}_{tot} >_l$ becomes independent of $l$ and which implies a finite correlation length of damage events. This suggests that the Maxwell–EB system progressively looses the memory of it's initial homogeneous, undamaged state and that an elasto-brittle material experiencing both healing and damaging enters a "marginally stable" state with scale invariance spanning the size of the system. This result is consistent with the scale-dependence analysis of RGPS-derived deformation rates of Marsan et al. (2004) and Stern and Lindsay (2009), in which no cutoff scale was observed for $l$ varying between 10 and 1000 km, suggesting that Arctic sea ice is most often in a near-critical state.

## 5.3 Intermittency

In this section we characterize the temporal behaviour of the Maxwell–EB model. Figure 7a represents the simulated macroscopic stress as a function of time (black dashed-dotted line) along with the corresponding damage rate (grey solid line) record for one realization of the uniaxial compression experiment with $N = 40$. Inspection of both temporal series reveals two types of mechanical behaviour of the Maxwell–EB material.

First, the evolution of the macroscopic stress is clearly characterized by cycles of slow stress build-ups and very fast relaxations. The strong asymmetry of the signal in time is confirmed by a high (negative) skewness ($-6$) of the distribution of the macroscopic stress increments $\frac{\Delta\sigma_m}{\Delta t}$ (not shown). Associated with these cycles is a succession of progressive increases in damage events and very sharp drops, after which damaging stops momentarily (red arrow on Fig. 7a).

Second, as identified on the same time series, some periods (e.g. the interval delimited by the dashed red box) are characterized by a continuous damage activity and by both low amplitude and low frequency fluctuations of the stress. This contrasted be-

**TCD**

doi:10.5194/tc-2015-200

**A Maxwell–Elasto-Brittle rheology for sea ice modelling**

V. Dansereau et al.

haviour translates into a significantly more symmetric (skewness of $-1.9$) distribution of $\frac{\Delta \sigma_m}{\Delta t}$. Inspection of the spatial distribution of damage (Fig. 7b) and strain rate fields (not shown) over this time interval indicates that the same system of interacting faults remains activated, with not much damaging activity over the rest of the domain and therefore suggests that creep-like deformation along this system dissipates all of the input loading.

Following the approach taken for fracture-type models which record the number of broken fibres, ruptured bounds, depinning events, etc., we investigate the time-dependence of the simulated damage activity by analyzing time series of the discrete failure events. We estimate the power spectral density (PSD) of damage rate time series. The resulting squared Fourier coefficients are averaged over 5 realizations of the compression experiment initialized with different fields of $C$ over domains with $N = 40$. Figure 8a represents the spectral density estimated by averaging the power over a 5 values window centred on each frequency $f$. We checked that using a smaller averaging window does not affect the shape of the PSD discussed below.

At low frequencies, the PSD is almost flat, suggesting that the number of damage events is uncorrelated in time. As these frequencies are lower than $\frac{1}{T_h}$, this is consistent with the fact that the Maxwell–EB material entirely looses the memory of previous damage events when allowed to heal completely. At higher frequencies, the PSD shows a decrease with increasing $f$ reminiscent of a temporal correlation of damaging events in the material. This expresses as a power law decay with $\mathrm{PSD}(f) = \frac{1}{f^\gamma}$. At intermediate frequencies, we estimate a slope $\gamma = 2$, suggesting that the instantaneous damage rate is correlated in time but *increments* of the damage rate are uncorrelated. At the highest frequencies, $\gamma > 2$, indicating that the damage rate is correlated in time and the of damage rate are anti-correlated. The break in the slope occurs around $f = 10^6$, a frequency that we relate to the minimum propagation time of a macro-rupture, i.e. the time of propagation of damage (i.e. of an elastic shear wave with speed $c$) across the width $\frac{l}{2}$ of the domain ($N$ time steps). The transition between the flat and power law decaying parts of the PSD is marked by a clear peak spanning the range of frequencies

Discussion Paper | Discussion Paper | Discussion Paper | Discussion Paper | Discussion Paper

**TCD**

doi:10.5194/tc-2015-200

**A Maxwell–Elasto-Brittle rheology for sea ice modelling**

V. Dansereau et al.

**TCD**

doi:10.5194/tc-2015-200

**A Maxwell–Elasto-
Brittle rheology for
sea ice modelling**

V. Dansereau et al.

corresponding to the cycles of healing and damaging, the red dashed line indicating the frequency of such a cycle, as identified by the double arrow on Fig. 7a.

Finally, we analyze the dependance of the simulated deformation on the time scale of observation using a temporal coarse-graining method (e.g. Rampal et al., 2008). Components of the strain rate at a given spatial scale are averaged over a time window of duration $t$ to compute the mean total deformation $< \dot{\varepsilon}_{tot} >_t$. The window is centred on an arbitrary time $t_0$ and has a size $t = 2n \times (N\Delta t)$ with $n = 1, 2, 3, ...$ and with the smallest averaging time scale corresponding to the time of propagation of an elastic shear wave with speed $c$ across the width $\frac{L}{2}$ of the domain. The chosen spatial averaging scale is that of the highest deformation rate, which as shown in Sect. 5.2 is of $\frac{L}{N}$. The domain is therefore divided in square boxes of equal size $l = \frac{L}{N}$ and the calculated deformation invariants are averaged over all available boxes. Figure 8b shows the total deformation rate $< \dot{\varepsilon}_{tot} >_t$ as a function of the time of observation $t$ (thick black line) averaged over 20 realizations of the coarse graining calculation (thin, coloured lines) centred on different $t_0$ for a simulation with $N = 40$. Consistent with the localizing of the deformation and an intermittent process, $< \dot{\varepsilon}_{tot} >_t$ decreases with increasing $t$ over almost two orders of magnitudes of $t$. The observed scaling is however altered in two ways, which relate to the specific geometry, loading and boundary conditions used in the present simulations. First, as one main fault always dominates the deformation in the system, curves of $< \dot{\varepsilon}_{tot} >_t$ are strongly modulated by a succession of peaks associated with the cycles of stress build-up and macro-rupture, the amplitude of which decreases with the scale of observation $t$. Second, at large $t$, the scaling assymptotes to a value corresponding to the prescribed forcing. Simulations over larger systems using non-homogeneous surface forcing should allow for multiple macroscopic scale faults to be active simultaneously and hence to observe a clearer scaling of the simulated deformation over larger time spans.

Discussion Paper | Discussion Paper | Discussion Paper | Discussion Paper | Discussion Paper |

**TCD**

doi:10.5194/tc-2015-200

**A Maxwell–Elasto-Brittle rheology for sea ice modelling**

V. Dansereau et al.

# 6 Conclusions

In this paper we have presented a new mechanical framework suited for modelling the brittle behaviour of the sea ice cover (Weiss et al., 2007) while keeping a continuum description. A relaxation term for the internal stress is added to the original Elasto-Brittle constitutive relationship and both the linear and viscous components are coupled to a progressive damage mechanism to allow partitioning between the reversible and permanent deformations within the material based on its local level of damage.

Highly idealized simulations using forcing conditions homogeneous in both space and time show the Maxwell–EB model simulates a complex temporal and spatial evolution of the deformation patterns, in close agreement with observations of the Arctic sea ice cover. Anisotropy in the simulated damage and deformation fields arises naturally from elastic interactions, although the material's properties are fully isotropic at the element scale. The model also reproduces both the persistence of creeping leads and the activation of new leads with different shapes and orientations, in agreement with the observed deformation of sea ice (Coon et al., 2007). Analyses of the simulated damage and deformation fields reveal

1. A highly heterogeneous deformation, translating into a power law decrease of the deformation rate with increasing spatial scale. The associated exponent varies periodically: it is highest in the vicinity of macro-rupture events and decreases between events as the material partially heals. The disappearance after a few "spinup" rupture events of a cross-over scale at which deformation rates become independent of the scale of observation suggests that the Maxwell–EB model, including both damaging and healing processes, successfully reproduces a "marginally stable" state, as observed for Arctic sea ice.

2. An intermittent deformation, manifested by the highly asymmetric temporal evolution of the internal stress within the material, which shows a succession of slow build-ups and very rapid relaxation phases. This intermittency is supported by the existence of a temporal correlation in the rate of damage at all timescales below

the material's characteristic healing time. A temporal scaling of the deformation rate is also obtained but due to the specific setup of the simulations analyzed here, it is modulated by the cycles of stress build-up and relaxation and its span is limited by the prescribed forcing.

Considering the highly idealized setup of the simulations analyzed here, these temporal and spatial scaling properties in the deformation fields cannot possibly be inherited from the prescribed forcing. Instead, their emergence is a signature of the mechanical behaviour of the Maxwell–EB model itself.

The next logical step in the development of a Maxwell–EB sea ice rheology consists in analyzing the sensitivity of the simulated deformation and damage fields to the model parameters. In particular, the partitioning between the simulated brittle and creep-like behaviour as well as the degree of localization of the deformation (Frederiksen and Braun, 2001) might depend on the rate of decrease of the viscous relaxation time with increasing level of damage (parameter $\alpha$) and on the characteristic time for healing and associated healing parameterization, all of which are poorly constrained in the case of the ice pack.

Further validation of the Maxwell–EB framework and the determination of the range of model parameters values suitable for sea ice call for a thorough comparison of the scaling properties of the simulated deformation rates with that estimated from the available ice buoy and RGPS data. Such analysis necessitates carrying numerical experiments over periods of several days to months and over realistic domains of regional to global scales. At these spatial and temporal scales, deformations within the sea ice cover become large. Hence advective processes cannot be neglected. As the Maxwell–EB rheology effectively reproduces very strong spatial gradients within the velocity, strain and stress fields, its use in large-deformation experiments requires the implementation of a robust advection scheme in order to limit diffusion and retain the strong localization of damage and deformation rates. The development of a numerical scheme for the the Maxwell–EB model that includes advection and is both efficient and prac-

**TCD**

doi:10.5194/tc-2015-200

**A Maxwell–Elasto-Brittle rheology for sea ice modelling**

V. Dansereau et al.

Discussion Paper | Discussion Paper | Discussion Paper | Discussion Paper

tical in view of dynamic-thermodynamic and fully coupled ocean-sea ice-atmosphere simulations is underway.

*Acknowledgements.* The financial support of TOTAL EP RECHERCHE DEVELOPPEMENT is gratefully acknowledged. V. Dansereau has been supported by ANRT. A. Audibert-Hayet, E. Coche and K. Riska are thanked for valuable suggestions and support on this work. V. Dansereau acknowledges support from the National Sciences and Engineering Research Council of Canada and from the Fonds Québécois de la Recherche sur la Nature et les Technologies.

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

**Table 1.** Model variables, parameters and domain dimensions for the uniaxial compression experiment.

| Parameters | | Values |
|---|---|---|
| Poisson's ratio | $\nu$ | 0.3 |
| Internal friction coefficient | $\mu$ | 0.7 |
| Ice density | $\rho$ | $900\,\mathrm{kg\,m^{-3}}$ |
| Shear wave propagation speed | $c$ | $500\,\mathrm{m\,s^{-1}}$ |
| Undamaged elastic modulus | $E^0$ | $2c^2(1+\nu)\rho\,\mathrm{Pa}$ |
| Undamaged apparent viscosity | $\eta^0$ | $10^7 \times E^0\,\mathrm{Pa\,s}$ |
| Minimum apparent viscosity | $\eta_{\mathrm{min}}$ | $10^4\,\mathrm{Pa\,s}$ |
| Cohesion | $C$ | $(25-50)\times10^3\,\mathrm{Pa}$ |
| Damage parameter | $\alpha$ | 4.0 |
| Undamaged relaxation time | $\lambda^0$ | $\frac{\eta^0}{E^0}\,\mathrm{s}$ |
| Characteristic time for damage | $t_{\mathrm{d}}$ | $\Delta t\,\mathrm{s}$ |
| Characteristic time for healing | $t_{\mathrm{h}}$ | $10^5\,\mathrm{s}$ |
| Dimensions of compression experiment | | Values |
| Length of the ice plate | $L$ | $200\times10^3\,\mathrm{m}$ |
| Prescribed velocity of forced edge | $U$ | $10^{-3}\,\mathrm{m\,s^{-1}}$ |
| Number of elements along short edge | $N$ | 10, 20, 40, 80, 100 |
| Mean model resolution | $\Delta x$ | $\frac{L}{2N}\,\mathrm{m}$ |
| Model time step | $\Delta t$ | $\frac{\Delta x}{c}\,\mathrm{s}$ |
| Ice thickness | $h$ | $1\,\mathrm{m}$ |
| Ice concentration | $A$ | $100\,\%$ |
| Variables | | Non-dimensional equivalent |
| Horizontal dimension | $x$ | $\tilde{x}=\frac{x}{L}$ |
| Time | $t$ | $\tilde{t}=\frac{tU}{L}$ |
| Ice velocity | $\boldsymbol{u}$ | $\tilde{\boldsymbol{u}}=\frac{\boldsymbol{u}}{U}$ |
| Internal stress | $\sigma$ | $\tilde{\sigma}=\frac{\sigma}{E^0}$ |
| Level of damage | $d$ | $\tilde{d}=d$ |
| Ice thickness | $h$ | $\tilde{h}=\frac{h}{H}$ |

TCD

doi:10.5194/tc-2015-200

**A Maxwell–Elasto-Brittle rheology for sea ice modelling**

V. Dansereau et al.

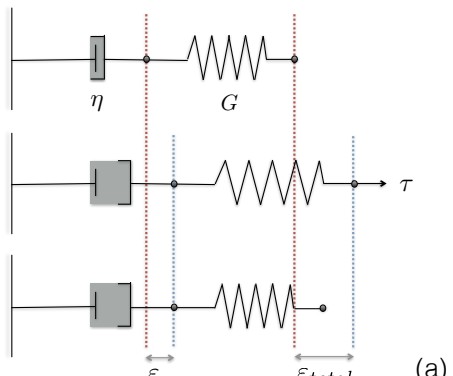

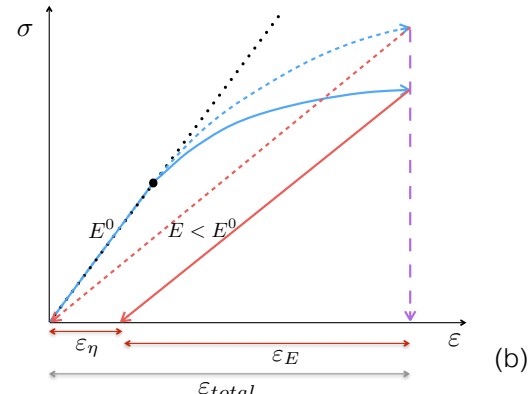

(a)

(b)

**Figure 1. (a)** Schematic representations of the Maxwell model for a continuum material with elastic (shear) modulus $G$ and viscosity $\eta$. At time $t$, a stress is applied on the system. It is removed at time $t + \Delta t$: the spring goes back to its initial position but the dashpot retains its deformation $\varepsilon_\eta$. **(b)** Loading-unloading paths for a material with initial elastic modulus $E^0$ in the linear-elastic (dotted), EB (dashed) and Maxwell–EB (solid lines) model. The black dot indicates the onset of damaging in the EB and Maxwell–EB models. Unlike the EB model, the Maxwell–EB model allows partitioning the total deformation into a permanent and an elastic contribution, indicated by the red arrows along the deformation axis. The diagram is not to scale in the context of modelling the lithosphere or sea ice: in these geomaterials, permanent deformations can become much greater than elastic deformations as damage events accumulate over time.

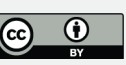

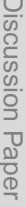

**TCD**

doi:10.5194/tc-2015-200

**A Maxwell–Elasto-Brittle rheology for sea ice modelling**

V. Dansereau et al.

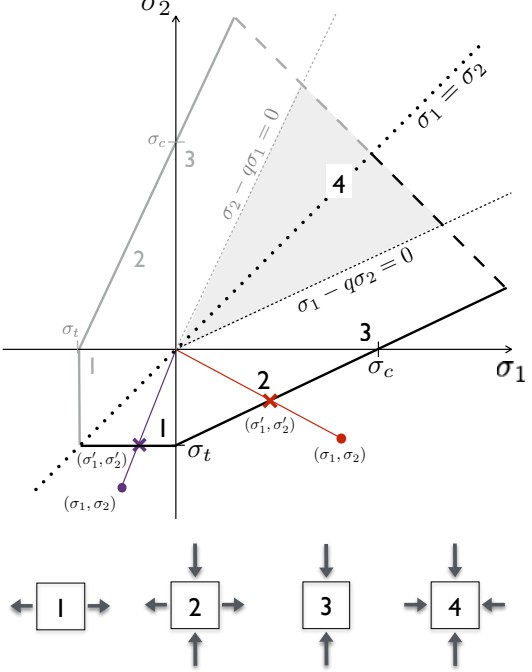

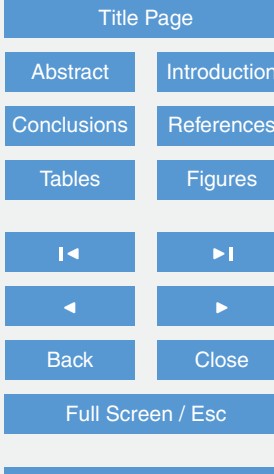
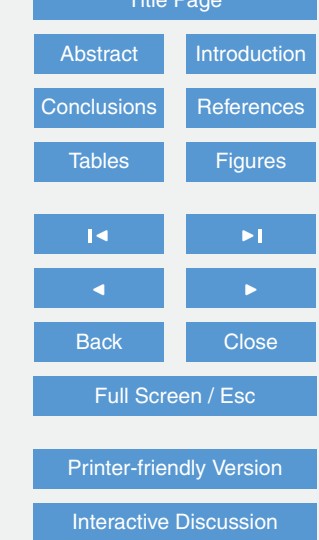

**Figure 2.** Damage criterion of the Maxwell–EB model in the principal stresses plane (solid line) combining the Mohr-Coulomb and tensile stress criteria. The thick dashed line represents a bi-axial compression truncation that closes the envelope but is not applied in the present model. Compression is taken positive and the dotted line indicates the $\sigma_1 = \sigma_2$ axis. Numbers indicate the states of (1) uniaxial tension, (2) biaxial tension and compression, (3) uniaxial compression and (4) biaxial compression and their location relative to the envelope. The calculation of the distance to the damage criterion $d_{\text{crit}}$, defined by the intersection $(\sigma_1', \sigma_2')$ of the line relating the state of stress $(\sigma_1, \sigma_2)$ of a given element to the origin of the principal stress plane, is represented in red in the case of exceeding the Mohr-Coulomb criterion and in purple, the tensile strength criterion.

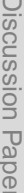

**TCD**

doi:10.5194/tc-2015-200

**A Maxwell–Elasto-Brittle rheology for sea ice modelling**

V. Dansereau et al.

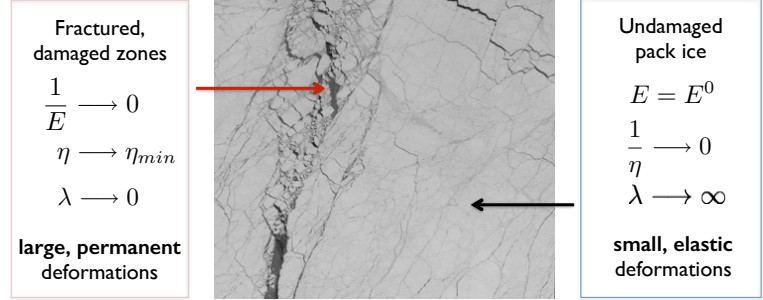

Fractured, damaged zones

$$\frac{1}{E} \longrightarrow 0$$

$$\eta \longrightarrow \eta_{min}$$

$$\lambda \longrightarrow 0$$

**large, permanent** deformations

Undamaged pack ice

$$E = E^0$$

$$\frac{1}{\eta} \longrightarrow 0$$

$$\lambda \longrightarrow \infty$$

**small, elastic** deformations

**Figure 3.** Dependence of the apparent viscosity ($\eta$) the elastic modulus ($E$) and the relaxation time ($\lambda$) on the level of damage in the Maxwell–EB sea ice model. The image is a SPOT satellite aerial picture of a 59 km × 59 km portion of the Arctic sea ice cover centred around 80.18° N, 108.55° W.

# TCD

doi:10.5194/tc-2015-200

**A Maxwell–Elasto-Brittle rheology for sea ice modelling**

V. Dansereau et al.

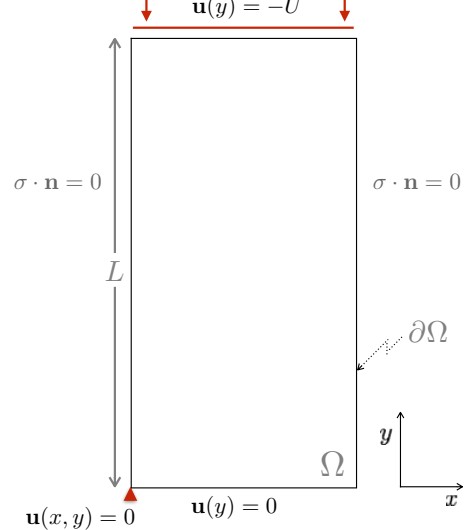

**Figure 4.** Domain and boundary conditions for the uniaxial compression experiment.

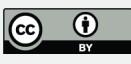

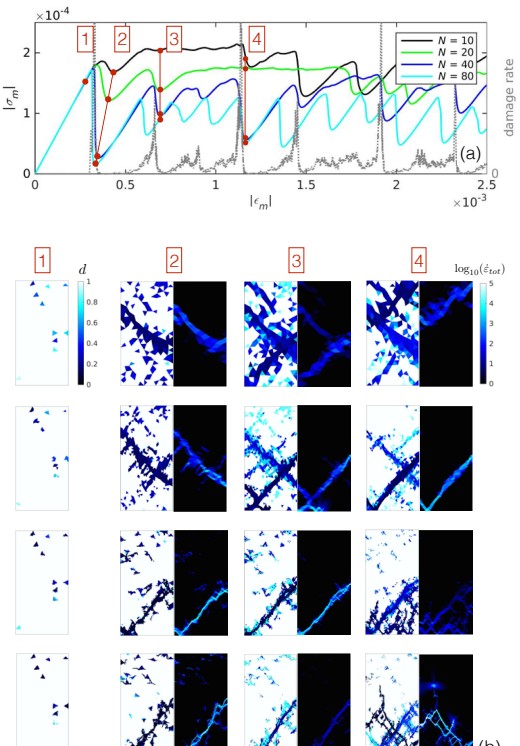

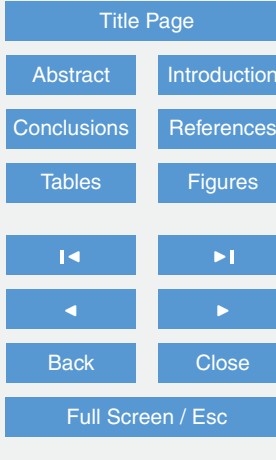

Discussion Paper | Discussion Paper | Discussion Paper | Discussion Paper |

**TCD**

doi:10.5194/tc-2015-200

**A Maxwell–Elasto-Brittle rheology for sea ice modelling**

V. Dansereau et al.

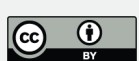

**Figure 5. (a)** Macroscopic stress vs. macroscopic strain (solid lines) for four uniaxial compression simulations with different spatial resolutions and damage rate (dashed grey line) for the simulation with $N = 40$. All simulations are initialized with the same values of mechanical parameters and cohesion field $C$ defined at the lowest spatial resolution ($N = 10$). **(b)** Fields of the instantaneous damage (left panels) and of the order of magnitude of the total deformation rate ($\log_{10}(\dot{\varepsilon}_{tot})$, right panels) at the four different times indicates on **(a)** and for the four simulations (resolution increasing from top to bottom).

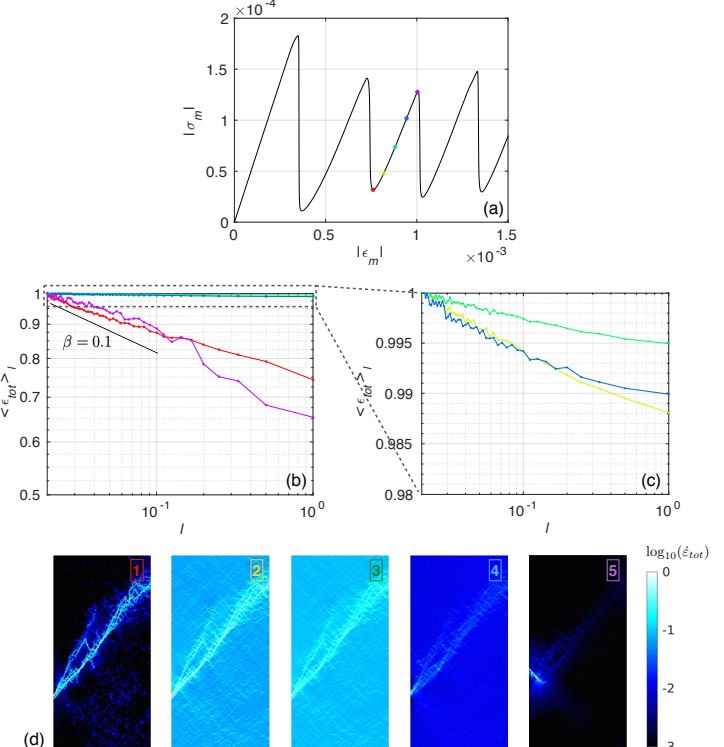

**Figure 6. (a)** Macroscopic stress as a function of the macroscopic strain for one realization of the uniaxial compression experiment with $N = 100$. **(b)** Total deformation rate as a function of the spatial scale $l$ ($l = \frac{L}{2n}$ with $1 \leq n \leq \frac{N}{2}$), normalized at the smallest scale $\frac{L}{N}$, at the five stages indicated on **(a)**. **(c)** Zoom into **(b)** for the second, third and fourth stages. **(d)** Corresponding fields of the order of magnitude of the total deformation rate ($\log_{10}(\dot{\varepsilon}_{tot})$) normalized by the maximum value of $\dot{\varepsilon}_{tot}$.

Discussion Paper | Discussion Paper | Discussion Paper | Discussion Paper

TCD

doi:10.5194/tc-2015-200

A Maxwell–Elasto-Brittle rheology for sea ice modelling

V. Dansereau et al.

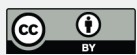

# TCD

doi:10.5194/tc-2015-200

**A Maxwell–Elasto-Brittle rheology for sea ice modelling**

V. Dansereau et al.

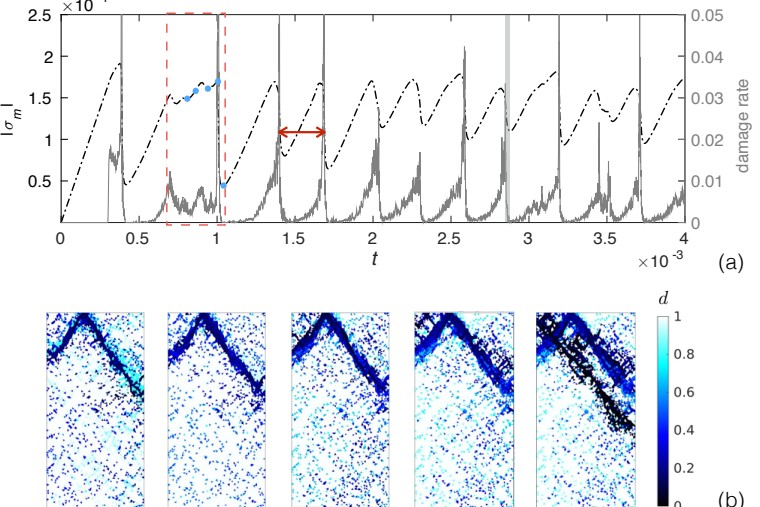

(a)

(b)

**Figure 7. (a)** Macroscopic stress (black dashed-dotted line) and damage rate (solid grey line) as a function of time for one realization of the uniaxial compression experiment with $N = 40$. The dashed red box indicates an interval of uninterrupted damaging activity, during which deformation is accommodated by a persisting system of interacting faults. **(b)** Instantaneous fields of level of damage at the five times indicated by blue dots on the macroscopic stress curve, showing the formation of the system of faults (first panel), which remains active for some time (three following panels), until the propagation of a new, non-interacting fault (last panel).

Discussion Paper | Discussion Paper | Discussion Paper | Discussion Paper

Discussion Paper | Discussion Paper | Discussion Paper | Discussion Paper |

**TCD**

doi:10.5194/tc-2015-200

**A Maxwell–Elasto-Brittle rheology for sea ice modelling**

V. Dansereau et al.

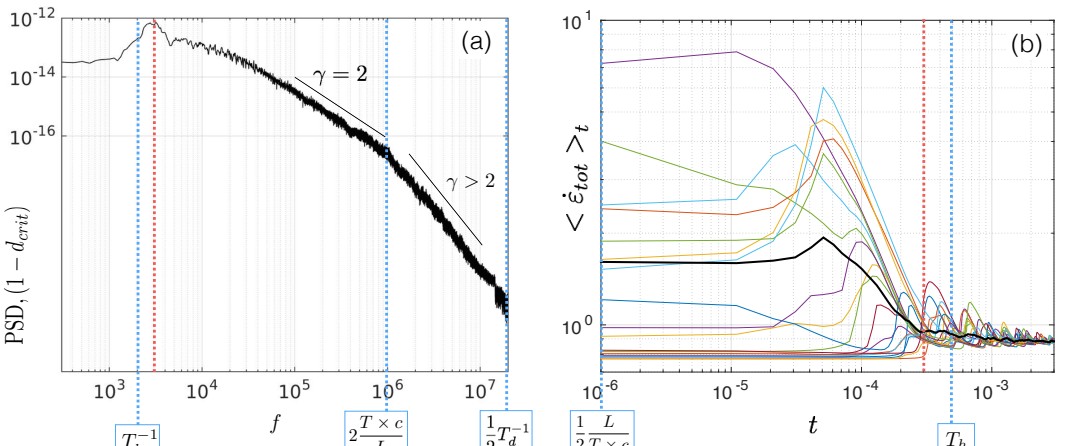

**Figure 8. (a)** Average power spectral density of the damage rate time series for 5 realizations of the uniaxial compression experiment initialized with different fields of $C$ and with $N = 40$. Blue dashed lines indicate, from left to right, the frequency associated with the characteristic time for healing, the inverse time of propagation of damage across the width of the domain and $\frac{1}{2} \times$ the frequency associated with the characteristic time for damage. The red dashed line indicates the frequency of the healing and damaging cycle marked with an arrow on Fig. 7a. **(b)** Total deformation rate $< \dot{\varepsilon}_{tot} >_t$ as a function of the observation time $t$, for 20 realizations of the coarse graining calculation centred on different arbitrary times $t_0$ along a uniaxial compression experiment with $N = 40$ (coloured lines) and average of the 20 realizations (thick black line).