# Peer review of "A Maxwell-Elasto-Brittle rheology for sea ice modelling"

_The Cryosphere, 2015_

## Referee Comment (RC1) · A. Keller (Referee) · 7 Mar 2016

**V. Danserau et al.: Maxwell-Elasto-Brittle rheology for sea ice modelling**

In this article, the authors present a viscoelastic damage model of sea ice mechanics. In contrast to existing models, this model takes into account transient elastic deformations and dynamic stress redistributions. Furthermore, a numerical solution of the model is applied to a simple test geometry, and the scaling behavior with respect to time and space discretizations is investigated. In my opinion, this is a very smart generalization of the modeling frameworks existing today. The concept is promising and deserves to be published.

Nevertheless, I think that the article still requires a number of changes and improvements in order to make it stand out. There are mainly three points which I am concerned about, and which the authors should revise/discuss more in detail:

The **literature discussion** in the introduction is very much focused on sea ice modeling. Damage mechanics has been a vital topic within the glacier and ice sheet modeling community in recent years though, and – even though length, time and stress scales may differ by several orders of magnitude – the models used in this context are conceptually not fundamentally different from those used for sea ice. Particularly, the pioneering work by Pralong et al. (2006) (and several other articles by the same authors) might be important; but also later contributions, such as Duddu and Waisman (2013), could potentially be relevant.

**Damage framework:** In the damage mechanics literature, a number of different damage measures have been proposed. They differ in how they are included in the constitutive framework and in their level of anisotropy. The damage evolution has to be prescribed depending on the choice of the damage measure. I think the authors should explain their choice in more detail! Why does $d_{crit}$, and thus the stress, enter the damage evolution equation linearly? The level of anisotropy present in a damage theory is reflected by the tensorial structure of the damage measure. In this light, using a scalar damage measure in a framework which is supposed to model induced anisotropy should carefully be justified. Furthermore, damage healing is known to be delicate with respect to the entropy principle (see also Pralong's article cited above). I am not sure to which extent this issue affects the healing parameterization given in this work, but at least I think it should be

addressed as a possible problem.

**Language:** As far as I can tell (English not being my first language), the language of the article frequently sounds a little clumsy, or at least significantly deviates from usual conventions of scientific English. I suggest it to be revised by a native speaker of English.

Apart from that, I would like to suggest a number of minor changes (which the authors are free to accept or not). A list of them is attached below.

Zurich, March 7, 2016                                                                 Arne Keller

**Suggested corrections/changes:**

- p.2 l.6 (and many other places): "constitutive relationship:" Even though admittedly a constitutive relationship sounds very romantic, I guess the more common terminology in this context is "constitutive relation."

- p.3 l.25: "physically consistent with observed behavior:" Is it physically consistent, or consistent with observed behavior, or both? In either case, that should be made clear.

- p.5 l.9: "processing... method:" I am not sure whether "processing" is the right word for this.

- p.5 l.11: "continuum solid:" Shouldn't this be a "continuous solid?"

- p.5 l.25-p6l2: Maybe make two sentences out of this in order to make it more readable.

- p.6 l.5: "material's velocity" rather "the velocity of the material" (as the material is not a person).

- p.6 l.13f: "the ration of ... and of the time": drop second "of"

- p.6 l.25: add commma after "However"

- p.7 l.1: "i.e. .. forcing" what do you mean by this?

- p.7 l.5 what does "such a model" refer to?

- p.7 l.20ff: I do not quite understand why the use of a viscoelastic constitutive relation has to be justified from rock mechanics? There is a large literature about viscoelasticity of ice.

- p.8 l.26ff: Even if I feel bad about this self promotion, but a very similar model for glacier ice has been proposed by Keller and Hutter (2014).

- p.9 l.23: I guess the strain rate tensor is the symmetric gradient of the velocity? Please state this explicitly. Furthermore, I think something should be said about how strain rates and strains are related (the notation suggests that the strain rate tensor is the rate of the strain tensor, which is generally not the case).

- p.10 l.20: Maybe write this as equation.

- p.11 l.4: The principal stresses are eigenvalues, not components.

- p.13 2nd paragraph: The physical interpretation of the damage variable crucially depends on how it affects the stress-strain relation. Therefore, I think this should be discussed right when defining the damage measure.

- p.13 l.22f: Please define $h$.

- p.14 l.22ff: "Time steps" and "elements" are concepts of numerical methods. One should be careful with using them for motivating the governing equations of a model. In physics, space and time are not discrete.

- p.15 l.1: Equation numbers missing? Please give a clear definition of $\epsilon$ (particularly in the context mentioned above, strains and strain rates).

- p.15 l.6: Again, principal stresses are not stress components (they do not have the transformation properties of tensor or vector components).

- p.17 l.10: It may be overly rigorous, but I think it is not a proper use of the Landau notation to write $\mathcal{O}$(some constant number)

- p.17 l.19: "modelling," vs. e.g. p.16 l.15, "parameterizations:" It is better style to consistently use either British or American spelling.

- p.18 l.3: "the the"

- p.18 l.23: "... are entirely defined by ... :" Either they are well-defined or not, there is nothing in between, so no need for this emphasis. Maybe better rephrase to "... only depend on"

- p.18 l.24: "the constitutive equation..." please specify which one (the constitutive framework consists of more than one equation).

- p.19 Eqs. 12-13: I think this regularization technique would be worth a more concise discussion: Is this purely a method to ensure convergence of the numerics? (In this case I'd suggest not to mix this with the physical governing equations). Or is this a conceptual problem? Finally, this is a known problem in continuum damage mechanics that the more damage is increasing the weaker its physical interpretation becomes.

- p.19 l.17f: "we take this approach.... but it had really no impact on our results.... :" This sounds very sloppy (and the switch from present to past tense is somewhat random).

- p.20 l.1: "a 2-dimensional plate ... and a constant healing rate ..." What are the consequences of the two-dimensional plate assumption? Probably a simpler velocity field? Apart from that, it sounds odd to me to squeeze those two assumptions (concerning completely different parts of the model) into one sentence.

- p.20 l.6: Not sure whether "assimilate" is the right word for this.

- p.20 l.9f: What does "internal stress" refer to, Cauchy stress? And shouldn't it rather be distributed "over" the depth? What is the advantage of keeping the entire stress tensor?

- p.20 l.14: Definition of the ice concentration?

- p.20 l.17ff: Rather make two sentences out of this.

- p.21 l.3: What is the reason not to write $A = 1$?

- p.21 l.6: "c*" is the * a typo?

- p.21 l.20f: "the dynamical system... read:" should be singular ("reads").

- p.22 l.11: "...parameters must evolve within..." so, they evolve while the model is running? This would change the dynamical equations. Otherwise, rather rephrase this.

- p.22 l.14: "characteristic time for damaging" Rather write "damage evolution" instead of "damaging" (idem in several other passages).

- p.22 l.23: typo "*One* the one hand"

- p.23 l.6: "propagation of the damage:" drop the article.

- p.23 l.13ff: "This separation.... calculations:" Somewhat weird semantics and ambiguous syntax in this sentence. The content absolutely makes sense though. Maybe rephrase this (and split into two sentences).

- p.23 l.17: "Considering the estimates ... aforementioned:" Either write "the aforementioned estimates" or "the estimates ... mentioned before."

- p.24 l.4: "over undamaged... areas:" rather "in" (idem l14).

- p.24 l.13: typo "euqation"

- p.24 l.15: "To get round this problem:" sounds very sloppy....

- p.24 l.21: ".. function, unnecessary:" no comma (maybe rather write "is unnecessary").

- p.25 l.5: "... transmitting the damage information within the material:" Not sure whether "within" is the appropriate preposition.

- p.25 l.6ff: I don't quite understand this logic... If the waves are not resolved, why should they be filtered out of the solution? Furthermore, replace "the model's solution" by "the solution of the model" (A model *can* be a person, just in this case for sure it is not ;-))

- p.25 l.25: Rather "*in* a material..."

- p.26 l.4: "time derivative for the Cauchy stress..." derivative *of*

- p.26/27, description of the test geometry: I don't quite understand the boundary conditions. Is the velocity on the lower (short) edge set to $(0,0)$ or may there be a non-zero component in $x$-direction? Concerning the lateral boundary, in the text it says "no confinement is applied on the lateral sides" (that is, the boundary may move freely?), whereas in the sketch (Fig. 4) it says $\mathbf{u}(x,y) = 0$, thus the velocity is fixed. Please clarify these ambiguities! Furthermore, if the lateral boundary *is* fixed, what happens to the inflowing ice mass if the ice thickness is kept constant?

- p.27 l.26f: "so that to be representative:" the conjunction "so that" should not be followed by an infinitive.

- p.28 l.1: drop the comma.

- p.28 l.10-26: Maybe move this to the literature discussion?

- p.29 l.8: "... is its strong anisotropy" would this finding not call for the use of a tensor damage variable? Please explain why a scalar damage model is sufficient.

- p.30 l.15 "over undamaged parts" rather "in undamaged parts?"

- p.30 l.19: "spatial distribution of the damage criteria" isn't the damage criterion the same everywhere? Only its parameters vary.

- p.32 l.6: "we use the output of strain rate fields from simulations..." isn't it rather the output of simulations, not that of strain rate fields?

- p.32 l.24: "....and clusters in space ..." This seems somehow syntactically lost in the sentence?

- p.34 l.9: "of the discrete failure events:" drop the article.

- p.34 l. 25: "... the of damage rate are anti-correlated" something missing here?

- p.35 l.6: "mean total deformation $\langle \dot{\epsilon}_{\text{tot}} \rangle_t$" that's a deformation rate, not a deformation.

- p.36 l.7: "permanent deformations *within the material*" There are no deformations outside of the material, so no need to state this.

- p.36 l.9 "show the Maxwell-EB model simulates...." maybe make the beginning of the subordinate clause clear by using "that". Or even better split into two sentences.

- p.36 l.26: "internal stress within the material" why not just "the Cauchy stress"? Or even "the stress?" I guess there are neither external stresses, nor stresses outside of the material....

- p. 37 l.1: "of the material" instead of "the material's"

- p.37 l.20: "carrying numerical experiments" rather "carrying *out*"

- p.39 l.20: typo "shear faaults"

- p.50, Fig. 6b,c: Excessive use of colored plots. Except for the yellow one, they all look more or less the same to me. I am probably not the only one who will have trouble distinguishing those plots: statistically, you can expect that about one out of ten male readers has a similar color vision deficiency. So it probably makes sense to use dash patterns instead, or to add plot labels.

- p.51, Fig. 7a (and various other figures): Are the units dimensionless? If so, this should be made clear, e.g. with the tilde notation used in the text.

**References**

Duddu, R. and H. Waisman, 2013: A nonlocal continuum damage mechanics approach to simulation of creep fracture in ice sheets. *Computational Mechanics* **51**(6), 961–974.

Keller, A. and K. Hutter, 2014: A viscoelastic damage model for polycrystalline ice, inspired by weibull-distributed fiber bundle models. part I: Constitutive models. *Continuum Mechanics and Thermodynamics*, 1–16.

Pralong, A., K. Hutter, and M. Funk, 2006: Anisotropic Damage Mechanics for Viscoelastic Ice. *Continuum Mechanics and Thermodynamics* **17**(5), 387–408 doi:10.1007/s00161-005-0002-5.

---

## Referee Comment (RC2) · C.P. Borstad (Referee) · 23 Mar 2016

This manuscript details the development of a new viscoelastic rheology for sea ice modelling that includes fracturing behaviour through the use of a continuum damage variable. A Maxwell type viscoelastic model is adopted, with the elastic and viscous terms coupled in series. A continuum damage variable is adopted for representing fractures in a continuum sense. The damage variable serves to decrease both the effective elastic modulus and the effective viscosity in the Maxwell model. The constitutive relation for developing and modifying the damage variable is based on a Mohr-Coulomb criterion that accounts for both principal stress components (in 2D plan-view) and different levels of ultimate strength in tension versus compression. A healing scheme for damage is introduced, governed by a characteristic timescale for damage healing. The resulting governing equation for damage evolution thus includes both damage production according to the current stress state and damage reduction (healing) through time. The effective elastic modulus is decreased according to the current level of damage in a similar manner as is commonly adopted in purely elastic damage models. The effective viscosity, however, is decreased according to a power law such that the viscous relaxation time of the material decreases with increasing damage.

The manuscript is incredibly detailed and generally very well written. The behaviour of the model is thoroughly explored using an idealized, uniaxial compression experiment. The model is capable of reproducing a number of features that are consistent with observations of sea ice deformation, including heterogeneous deformation that localizes along faults/leads and intermittent deformation and damage at different spatial and temporal scales. Care is taken to point out the limitations of the model setup and directions for future development.

Most of my comments are relatively minor, and mainly have to do with the presentation and description of the model and the results.

**General comments**

1. The thoroughness and detail of the manuscript are commendable, although in some places the presentation was somewhat confusing as a result. I would recommend splitting the Introduction into separate Introduction and Background/Theory sections. A shorter Introduction section that clearly outlines the motivation for the new model, the most relevant context, and the general approach would benefit the reader. As it is written, the Introduction currently is very long and detailed but in some places confusing in terms of both the writing and the relevance of this level of detail. In other places, relevant details seem to be left out, and it seems to be left to the reader to be familiar with all of the references in order to understand certain points. It seems to be an excellent review of the state of sea ice modelling, and demonstrates that the authors have a good grasp of the field, but as a reader I found myself a little "lost in the weeds" at times.

2. Even in a fully viscous model of ice deformation, the stress balance can be non-local (for instance, in a glaciology context, viscous ice stream or ice shelf deformation is described by a stress balance that is inherently nonlocal, (e.g. *MacAyeal*, 1989)). You seem to be implying in several places that "long-range" interactions must come from elastic deformation (e.g. p 4, l 14). Am I reading this wrong, or are you indeed stating that long-range interactions can only be accounted for by elastic interactions?

3. The results of the model are mesh-dependent, as damage localizes to the scale of an individual element. This is often viewed as a negative result, because the results of the model thus depend on how the user sets it up. However, many different approaches for nonlocal regularization of damage models have been proposed and adopted in a variety of settings (e.g. *Bažant and Jirásek*, 2002; *Borstad and McClung*, 2011). In these approaches, the stresses/strains/constitutive relation are computed by integrating over an intrinsic length scale related to the scale of heterogeneity of fracturing of the material. As long as the element size is smaller than this intrinsic length scale, the results of the model are independent of

the resolution of the mesh. I think the authors should mention this type of approach in the manuscript, and discuss whether it might be feasible to produce mesh-objective results.

4. The discussion of the damage formulation is a bit confusing. You mention in the text (p 14, l 24-26) that stresses outside the failure envelope are non-physical. However, unless I am missing something, you seem to be calculating your damage variable according to the distance *beyond* the failure envelope. It seems, then, that damage is a sort of constitutive post-processing to "correct" the stress level such that it lies directly on the failure envelope. Some clarification is needed in the text on this point, since your schematic representation of the failure envelope in Figure 2 seems to contradict what you state in the text. Damage is based on the distance of the stress state outside of the envelope, and yet a stress state outside the envelope is non-physical...

5. I'm confused as to why a separate term for the ice concentration ($A$) is needed, as this seems at least partially redundant with damage. Why is it necessary to have both a concentration term and a damage term that modify both the elastic modulus and the viscosity? Isn't there some redundancy here, as a damaged fault/lead will necessarily have a reduced ice concentration? The ice concentration term seems to be simply added to the model at the very end of the model discussion, without much explanation.

**Specific comments**

- l 25: "or"

- p 4, l 6-9: how have these VP hypotheses been found inconsisent? Can you summarize these for the reader?

- check English spelling throughout the document, a number of words are misspelled (looses instead of loses, euqation, it's instead of its, dependance, assymptotes,...)

- p 10, l 20: there is some inconsistency in the text formatting of the different versions of "$I$" for the identity tensors

- p 13, l 22-23: I was confused about what $h$ is here

- p 19, l 18: might there be other contexts or model setups (e.g. realistic domains) for which the minimum value of $d$ might come into play? The results of a damage model can be quite sensitive to this choice.

- p 20, l 11-13: Some motivation or explanation is needed for why you choose to write the momentum equation in terms of internal stress rather than the vertically integrated stress tensor, especially if you are departing from what is more commonly done in the sea ice modelling community.

- p 23, l 6-9: why not perform a sensitivity analysis as you describe then?

- p 26, l 5: you previously described the inertial term as being negligible, so why is it here? some clarification is needed.

- The first part of the Results section is not really results, but background.

- p 29, l 23-24: this would be helpful to state also in the figure caption

- p 30, l 26-27: well, the localization scale is the element scale, so the choice of resolution dictates the localization of damage

- p 31, l 6-11: but you didn't introduce disorder initially, so you cannot claim this here.

- p 31, l 27: "...has already been investigated in depth..." is another example of the reliance on the reader to be familiar with all of the literature you are citing. It would be more helpful to summarize the findings. What did these investigations find?

- p 34, l 25: some word has been omitted here

- Figure 5: it doesn't look like the elements are getting smaller from the top row to the bottom row of panel (b), but isn't the resolution supposed to be getting finer moving down in the figure? Also, the damage rate axis in panel (a) is missing a numerical scale other than the zero.

- Figure 6 is presented in the discussion of heterogeneity, the dependence on the spatial scale of observation. It's still not clear to me how this is represented in the figure, which only shows one realization of the experiment at one resolution.

**References**

Bažant, Z. P., and M. Jirásek (2002), Nonlocal integral formulations of plasticity and damage: Survey of progress, *J. Eng. Mech. - ASCE*, *128*(11), 1119–1149, doi:10.1061/(ASCE)0733-9399(2002)128:11(1119).

Borstad, C. P., and D. M. McClung (2011), Numerical modeling of tensile fracture initiation and propagation in snow slabs using nonlocal damage mechanics, *Cold Reg. Sci. Technol.*, *69*, 145–155, doi:10.1016/j.coldregions.2011.09.010.

MacAyeal, D. (1989), Large-scale ice flow over a viscous basal sediment: Theory and application to Ice Stream B, Antarctica, *J. Geophys. Res.*, *94*(B4), 4071–4087.

---

## Author Comment (AC1) · 22 Apr 2016

**A Maxwell-Elasto-Brittle rheology for sea ice modelling**

Véronique Dansereau, Jérôme Weiss, Pierre Saramito and Philippe Lattes

**Answers to reviewer 1**

**GENERAL COMMENTS**

**The literature discussion** in the introduction is very much focused on sea ice modeling. Damage mechanics has been a vital topic within the glacier and ice sheet modeling community in recent years though, and – even though length, time and stress scales may differ by several orders of magnitude – the models used in this context are conceptually not fundamentally different from those used for sea ice. Particularly, the pioneering work by Pralong et al. (2006) (and several other articles by the same authors) might be important; but also later contributions, such as Duddu and Waisman (2013), could potentially be relevant.

The introduction is indeed focussed on sea ice modelling. However, as the Maxwell-EB model is developed to this specific purpose, we believe this focus is relevant. In particular, discussing the current state of sea ice models in terms of their capability to reproduce sea ice deformation appears essentials. The need for an improved representation of deformation has been discussed extensively in late years: we think it is important to recall this point. As we suggest a *new* mechanical framework for these models, we believe it is also important to state the motivations for our approach, and the fact that we were inspired by the known similarities between the brittle mechanical behaviour of sea ice and the Earth crust, hence by some of the methods used for modelling faults.

Viscoelastic models are indeed used for glacier ice. However, fundamental differences (at least 4) between the present viscoelastic model for sea ice (as well as models for the Earth crust) and viscoelastic models for (polycrystalline) ice and glaciers exist. You are right when stating that time scales, in particular, are key here. Differences in time scales translate into intrinsic differences in the nature of the deformation of sea ice versus of glacier ice: sea ice deforms "rapidly" under the action of the wind and ocean drags, in the *brittle* regime, while glaciers deforms slowly, through viscoplastic deformation of bulk ice.

In the case of viscoelastic models for sea ice and the Earth crust, *instantaneous* elastic deformations are *not* neglected and the viscosity associated with the relaxation of the stress is *not* the dynamical viscosity associated with "true" creep, but rather an *apparent* viscosity, intended to represent the large deformations within the fractured ice cover or faults. This last point is perhaps the most important difference between sea ice- and glacier-type models and in order for the reader to understand the need for an *apparent* rather than the *true* viscosity to model the deformation of sea ice, we believe the comparison to models of the Earth crust is relevant. Finally, the damage criterion in the present and previous (eg., EB) sea ice model is a function of a critical *stress*. In models for glacier flows, damage is often a function of a given cumulative *deformation* threshold and it impacts the viscous flow through the concept of effective stress on Glen's law.

However, we agree that these relevant distinctions deserves a few words and hence we included a short discussion of viscoelastic, damage-based glacier models and their references in the Background (former Introduction) section. To keep this section reasonably short, we somewhat shortened the discussion of models of lithospheric faulting (lines 13-18, page 8).

**Damage framework** In the damage mechanics literature, a number of different damage measures have been proposed. They differ in how they are included in the constitutive framework and in their level of anisotropy. The damage evolution has to be prescribed depending on the choice of the damage measure. I think the authors should explain their choice in more detail! Why does $d_{crit}$, and thus the stress, enter the damage evolution equation linearly? The level of anisotropy present in a damage theory is reflected by the tensorial structure of the damage measure. In this light, using a scalar damage measure in a framework which is supposed to model induced anisotropy should carefully be justified. Furthermore, damage healing is known to be delicate with respect to the entropy principle (see also Pralong's article cited above). I am not sure to which extent this issue affects the healing parameterization given in this work, but at least I think it should be addressed as a possible problem.

Different approaches have indeed been used for the damage mechanism. In absence of physical evidences for higher levels of complexity, in developing the Maxwell-EB model we sought the simplest possible formulation and based our approach on isotropic, progressive damage models involving a scalar level of damage variable (e.g., *Amitrano, 1999*, and the EB sea ice model of Girard et al., *2011*). In these models, the decrement in damage enters the damage formulation "linearly". The

main difference in the Maxwell-EB framework is that this decrement in damage, called $d_{crit}$, is not an *arbitrary constant* as in these previous progressive damage frameworks, which are based on a sub-iteration loop in which damage is allowed to evolve and stress to be redistributed from over to sub-critical elements until a steady-state is reached (i.e., until all states of stress lie within the yield envelope) before the model is incremented and loading resumes. In the present model, damage evolves in "real" model time. Hence if $d_{crit}$ was set constant, the stress drop associated with damaging can be such that stresses will still lie outside of the yield envelope after one model time step. As overcritical stresses are not physical, here we make the logical assumption that the decrement of damage $d_{crit}$ associated with a local damage event should be such that the value of the stress be at most equal to the local critical value.

Concerning anisotropy, we thank the referee for this important point, which was not detailed enough in the initial version of the manuscript. The key point is that, in the present model, an anisotropic damage formulation is not required to generate anisotropy of stress and strain fields. Indeed, in an elastic medium submitted to a non-perfectly isotropic loading (i.e., non-perfectly isotropic with respect to the domain geometry or the heterogeneity present in the material), the elastic kernel associated with a damaged "inclusion" (*Eshelby, 1957*) is anisotropic, hence is the redistribution of stresses. Therefore, the combination of (1) small-scale disorder, (2) damage mechanics in an elastic medium, and (3) this anisotropy of the elastic kernel itself is sufficient to generate anisotropy up to *very large* space scales through successive elastic interactions between damaged elements. We now discuss this point in more details in the *Results* section and add a figure (6) that shows the anisotropic perturbation in the (Coulomb) stress field ($\sigma_1$ - q$\sigma_2$) generated when uniaxial compression is applied to a uniform rectangular plate with an isotropic, circular inclusion and, similarly, when disorder is introduced in the model at the element scale.

Finally, concerning the entropy principle and the healing parameterization, you are right that this remains an open question. As mentioned in section 4.3.2, we used the simplest possible parameterization for healing (using a constant healing rate) in the present uncoupled implementation of the Maxwell-EB model. We did not verify the validity of our approach with respect to entropy. In a dynamic-thermodynamic sea ice model, the rate of healing should logically be a function of the local difference between the temperature of the air near the surface of the ice and the freezing point of seawater below (see section 4.3.2). However we believe entropy is hard, perhaps impossible, to quantify and monitor in the context of an *open*, dynamically and thermodynamically coupled system such as the Arctic Ocean.

**MINOR COMMENTS**

p.2 l.6 (and many other places): "constitutive relationship:" Even though admittedly a constitutive relationship sounds very romantic, I guess the more common terminology in this context is "constitutive relation."
We replace this formulation by "constitutive law".

p.3 l.25 "physically consistent with observed behavior:" Is it physically consistent, or consistent with observed behavior, or both? In either case, that should be made clear.
It is naturally both, as something that is physically consistent should necessarily be consistent with observation. We replace this sentence by: "consistent with its observed mechanical behaviour".

p. 5 l.9 "processing... method:" I am not sure whether "processing" is the right word for this.
"Processing" is synonym of "treating" here, or "representing". We rephrase: thereby treating discontinuum mechanics with a continuum mechanics method.

p. 5 l.11 "continuum solid:" Shouldn't this be a "continuous solid?"
OK.

p.5 l.25-p.6 l.2: Maybe make two sentences out of this in order to make it more readable.
OK.

p.6 l.5: "material's velocity" rather "the velocity of the material" (as the material is not a person).
OK.

p.6 l.13f: "the ration of ... and of the time": drop second "of"
OK.

p.6 l.25: add comm**m**a after "However"
OK.

p.7 l.1: "i.e. .. forcing" what do you mean by this?
Here we make the distinction between the part of the intermittency that is attributable to the forcing applied on the material and the part that is inherited from its mechanical behaviour. This is an important difference in the context of sea ice. The turbulent wind forcing itself exhibits some intermittency, which is "transmitted" to the ice cover. This part of the intermittency is therefore expected to be reproduced in most sea ice models. However, it was shown that the wind forcing is less intermittent than the deformation of sea ice, hence that the "extra" intermittency of sea ice deformation is attributable to its mechanical behaviour (*Weiss, 2008; 2013*). It is this part of the intermittency of the deformation that cannot be adequately reproduced in sea ice model if elastic interactions are not accounted for and if the memory of elastic stresses is not adequately retained.

p.7 l.5 what does "such a model" refer to?
It refers to the previous sentence: to a rheological model that has "the capacity to distinguish between reversible and irreversible deformations" and that "allows a passage between the small/elastic and large/permanent deformations". We try improving the reading by adding "that" after "such a model".

p.7 l.20: I do not quite understand why the use of a viscoelastic constitutive relation has to be justified from rock mechanics? There is a large literature about viscoelasticity of ice.
Once again, here we do not mean to model the classical bulk viscoelasticity of ice (see response to major comment above). Viscous creep is negligible in the deformation of sea ice, which is essentially brittle (e.g., *Weiss et al., 2007*). The apparent viscosity introduced here is intended to represent the slow relaxation of elastic stresses through permanent, large deformations of the damaged material: this is why we make a parallel with models of lithospheric faulting rather than with viscoelastic models for glacier ice.

p.8 l.26: Even if I feel bad about this self promotion, but a very similar model for glacier ice has been proposed by Keller and Hutter (2014).
See above.

p.9 l.23: I guess the strain rate tensor is the symmetric gradient of the velocity? Please state this explicitly. Furthermore, I think something should be said about how strain rates and strains are related (the notation suggests that the strain rate tensor is the rate of the strain tensor, which is generally not the case).
Yes, the strain rate tensor is the rate of the strain tensor and hence is the symmetric gradient of the velocity. We clarify this point by defining the strain rate tensor in equation (1) as the symmetric gradient of the velocity. *Large deformations* in the Maxwell-EB model are accounted for by introducing the objective derivative of the stress tensor in the constitutive law (eqn. 1). To clarify this point further, we include the definition of the objective material derivative for the stress tensor after introducing the constitutive relation and develop the $\beta_a$ term, which accounts for the effects of rotation. It is important to recall however that all simulations presented in the paper are in the small-deformation regime (no advection and rotational effects neglected), hence these distinctions do not apply here.

Besides, we recognize that the notations used to describe the standard Maxwell model at the beginning of section 4.1 were rather confusing, especially when referring to figure 1(a) and (b) to introduce these concepts. For instance, *G* was used for the elastic modulus and $\tau$ was used for the stress tensor in the text and in figure 1a, instead of *E* and $\sigma$ in figure 1b. Hence we slightly reformulated the description of the standard Maxwell model and changed the notation in figure 1(a). We believe this improves the reading and presents the transition from the standard to the Maxwell-EB model more clearly.

p.10 l.20: Maybe write this as equation.
This definition of the (adimensional) elastic stiffness matrix is quite standard. Hence we do not think this ought to be listed as a separate equation. But we reformulate this definition in index notation, which makes it somewhat easier to grasp.

p.11 l.4: The principal stresses are eigenvalues, not components.
OK.

p.13 2nd paragraph: The physical interpretation of the damage variable crucially depends on how it affects the stress-strain relation. Therefore, I think this should be discussed right when defining the damage measure.

We are not sure we understand this comment. On the one hand, in the elastic regime, the effective stress is given by $\frac{\sigma}{d}=E^0\varepsilon$ with $0<d\leqslant 1$, which is consistent with the typical interpretation of damage in progressive damage models for elastic materials. On the other hand, in the viscous regime, the effective stress is given by $\frac{\sigma}{d^\alpha}=\eta^0\dot{\varepsilon}$ which does contrast with the classical definition of the effective stress due to the introduction of the damage parameter (exponent $\alpha$). This viscous term does not represent a true viscous flow of the bulk material, but instead an apparent viscosity aimed at slowly relaxing the elastic stresses within a damaged material. Hence the notion of effective stress is not entirely relevant here. As the parameter α is larger than 1, our formulation means that damage plays a stronger role on the apparent viscosity than the effective stress concept would do. We agree that this point might have been confusing, especially as we state about the damage variable "This variable is interpreted as a measure of sub-grid cell defects or crack density (*Kemeny and Cook, 1986*) and is allowed to evolve (...)" (p. 15, lines 14-16). Hence we reformulate this passage as "This variable is interpreted as a measure of sub-grid cell defects".

p.13 l.22f: Please define *h*.
Thanks for catching this: *h* stands for the ice thickness.

p.14 l.22: "Time steps" and "elements" are concepts of numerical methods. One should be careful with using them for motivating the governing equations of a model. In physics, space and time are not discrete.
Here the space and time discretizations do not "motivate" the governing equations of the model, although they need to be taken into consideration when writing the continuous form of the damage equation (that is, the part of the evolution equation for *d* pertaining to the damaging process). As already discussed in this section, this arises because of our treatment of the damage mechanism, which is similar to that of linear-elastic progressive damage models (e.g *Amitrano, 1999*). The governing equation for this discrete process is written as a recursive relation (formerly on line 20, p. 15 and now numbered).
In the linear-elastic damage mechanics model on which the Maxwell-EB model is based, time does not enter the governing equations (e.g., *Amitrano, 1999*, and *Girard, 2010*), hence the formulation of an evolution equation for this mechanism is not an issue and the damage equation is simply written in this recursive form. Here evolution equations are written and as the damage process is tied to the space and time discretizations, the order of the time scheme and spatial and temporal resolutions need to be taken into account when writing the recursive equation *in continuous form*. We agree that this might constitute a limitation of the current Maxwell-EB model, especially as it requires the use of an explicit time-stepping scheme for the damage evolution and the use of a small model time step.

p.15 l.1: Equation numbers missing? Please give a clear definition of epsilon (particularly in the context mentioned above, strains and strain rates).
OK, we numbered this equation and defined epsilon, the *strain* tensor.

p.15 l.6: Again, principal stresses are not stress components (they do not have the transformation properties of tensor or vector components).
OK.

p.17 l.10: It may be overly rigorous, but I think it is not a proper use of the Landau notation to write O(some constant number)
We rewrite these formulations in words instead.

p.17 l.19: "modelling," vs. e.g. p.16 l.15, "parameterizations:" It is better style to consistently use either British or American spelling.
As the first author is Canadian, Canadian english conventions were used (which combines that of American and British spelling).

p.18 l.3: "the the"
OK.

p.18 l.23: "... are entirely defined by ... :" Either they are well-defined or not, there is nothing in between, so no need for this emphasis. Maybe better rephrase to "... only depend on"
OK.

p.18 l.24: "the constitutive equation..." please specify which one (the constitutive framework consists of more than one equation).
OK.

p.19 Eqs. 12-13: I think this regularization technique would be worth a more concise discussion: Is this purely a method to ensure convergence of the numerics? (In this case I'd suggest not to mix this with the physical governing equations). Or is this a conceptual problem? Finally, this is a known problem in continuum damage mechanics that the more damage is increasing the weaker its physical interpretation becomes.
As mentioned on lines 13 to 15, p. 19, this formulation is introduced to ensure mathematical consistency, i.e., so that the constitutive equation be defined in the limit of $d = 0$. It is not meant to handle the physical interpretation of damage in the limit of a "completely damaged" material. We mentioned this point with the intend of being as rigorous as possible, as it is how the damage equation is written in our numerical scheme. However, as also pointed out, the use of a "regularization technique" for $d = 0$, as well of this specific technique (as opposed to introducing a minimum value on $d$ instead of on $\eta$) has no impact on our results, as the level of damage $d$ never reaches zero values (we set $d^0 > 0$ in all simulations). Hence we agree that introducing this level of precision within the description of the governing equations might not be necessary in the present paper. We suggest to remove this entire paragraph. To take care of any ambiguities, we specify the condition $0 < d^0 = d(t = 0) \leq 1$ in the equations for $E$ and $\eta$ as a function of $d$ (end of former page 18). We modify figure 3 accordingly and reformulate lines 13 and 14, p. 13, as "The level of damage is equal to 1 for an undamaged material and *approaches* the value of 0 in the case of a "completely damaged" material."

p.19 l.17f: "we take this approach.... but it had really no impact on our results.... :" This sounds very sloppy (and the switch from present to past tense is somewhat random).
This sentence was cut (see above).

p.20 l.1: "a 2-dimensional plate ... and a constant healing rate ..." What are the consequences of the two-dimensional plate assumption? Probably a simpler velocity field? Apart from that, it sounds odd to me to squeeze those two assumptions (concerning completely different parts of the model) into one sentence.
The 2-dimensional plate assumption here is equivalent to the plane-stress assumption, in the sense that we assume no stresses in the z direction ($\sigma_{13}$, $\sigma_{31}$, $\sigma_{23}$, $\sigma_{32}$, $\sigma_{33}$ = 0). In this case, the (adimensional) elastic stiffness tensor $\boldsymbol{K}$ is defined such that for all symmetric tensor

$$\boldsymbol{\varepsilon} = \varepsilon_{ij} \, \forall \, i,j \, ; 1 \leq i,j \leq 2, \boldsymbol{K} : \varepsilon_{ij} = \frac{\nu}{1-\nu^2} \varepsilon \delta_{ij} + 2 \frac{1}{2(1+\nu)} \varepsilon_{ij}.$$

However, one difference is that, as in all regional and global sea ice models, the *z-components* of the deformation are not taken into account here (the sea ice momentum balance equation is 2-dimensional). Hence yes, the velocity field is calculated in the horizontal plane ($u_x$, $u_y$) only.
This assumption was listed independently of the constant healing rate assumption. We reformulate this sentence as: "As in regional and global sea ice models, the ice cover is considered as a 2-dimensional plate due to its very large aspect ratio and plane stresses are assumed. A constant healing rate is used".

p.20 l.6: Not sure whether "assimilate" is the right word for this.
It is now replaced by "represents".

p.20 l.9f: What does "internal stress" refer to, Cauchy stress? And shouldn't it rather be distributed "over" the depth? What is the advantage of keeping the entire stress tensor?
Yes, the internal stress refers to the in-plane stress, $\sigma$. In virtually all sea ice models based on the continuum assumption, it is the common name for the stress arising from the sum of the mechanical interactions between ice floes (*Weiss, 2013*, *Feltham, 2008*, and many others). It is thereby distinguished from *external stresses*, which is the formulation used for the skin (air and ocean) drags per unit area on the ice cover.
Here, the entire stress tensor instead of the deviatoric part is used, as the simulated material is a compressible, elastic solid.

p.20 l.14: Definition of the ice concentration?
The ice concentration is the fraction of a model grid cell covered by ice, or in other words, the surface of ice (as opposed to open water) per unit area. In the present model, it has the same definition as in virtually all continuum sea ice models developed since 1979 (i.e., since the Hibler viscous-plastic model). Hence we do not believe a lengthly definition of this variable is necessary.

p.20 l.17: Rather make two sentences out of this.
OK.

p.21 l.3: What is the reason not to write A = 1?
OK.

p.21 l.6: "c*" is the * a typo?
No, we did intent to use this symbol. This constant is usually denoted $C$ (*Hibler, 1979*). We use $c*$ to distinguish it from the cohesion in the Maxwell-EB and EB frameworks.

p.21 l.20f: "the dynamical system... read:" should be singular ("reads").
OK.

p.22 l.11: "...parameters must evolve within..." so, they evolve while the model is running? This would change the dynamical equations. Otherwise, rather rephrase this.
We rephrase: "In order for the Maxwell-EB model to represent the intended physics, the value of these parameters must lie within a certain range".

p.22 l.14: "characteristic time for damaging" Rather write "damage evolution" instead of "damaging" (idem in several other passages).
OK.

p.22 l.23: typo "One the one hand"
OK.

p.23 l.6: "propagation of the damage:" drop the article.
OK.

p.23 l.13: "This separation.... calculations:" Somewhat weird semantics and ambiguous syntax in this sentence. The content absolutely makes sense though. Maybe rephrase this (and split into two sentences).
Agreed. The sentence has been split and reformulated as
"This separation of scales ensures that elements cannot recover by healing more strength than they have lost by damaging within one time step. In the case of sea ice for instance, excess healing would effectively entail a net growth, or thickening, of the pack, a process that should instead be accounted for by thermodynamic balance considerations. However, considering the estimates of the speed of elastic waves and of the healing rate of leads aforementioned, the sea ice cover naturally meets the condition $T_h \ll T_d$".

p.23 l.17: "Considering the estimates ... aforementioned:" Either write "the aforementioned estimates" or "the estimates ... mentioned before."
OK.

p.24 l.4: "over undamaged... areas:" rather "in" (idem l14).
OK.

p.24 l.13: typo "euqation"
OK.

p.24 l.15: "To get round this problem:" sounds very sloppy....
Replaced by "this issue can be dealt with by (...)"

p.24 l.21: ".. function, unnecessary:" no comma (maybe rather write "is unnecessary").
OK.

p.25 l.5: "... transmitting the damage information within the material:" Not sure whether "within" is the appropriate preposition.
We do think "within" works here.

p.25 l.6: I don't quite understand this logic... If the waves are not resolved, why should they be filtered out of the solution? Furthermore, replace "the model's solution" by "the solution of the model" (A model can be a person, just in this case for

sure it is not ;-))

Agreed: this paragraph is somewhat confusing, and not strictly necessary. Hence we cut these two sentences. The main point here is expressed in the two remaining sentences of this paragraph, which is that inertial effects and the effect of the propagation of viscoelastic waves on the stress and deformation fields can be safely neglected in most sea ice implementations of the model.

p.25 l.25: Rather "in a material..."
OK.

p.26 l.4: "time derivative for the Cauchy stress..." derivative of
OK.

p.26/27, description of the test geometry: I don't quite understand the boundary conditions. Is the velocity on the lower (short) edge set to (0, 0) or may there be a non-zero component in x-direction? Concerning the lateral boundary, in the text it says "no confinement is applied on the lateral sides" (that is, the boundary may move freely?), whereas in the sketch (Fig. 4) it says u(x,y) = 0, thus the velocity is fixed. Please clarify these ambiguities! Furthermore, if the lateral boundary is fixed, what happens to the inflowing ice mass if the ice thickness is kept constant?

The figure that was included in this submitted version of the paper was not the good one, and yes, was somewhat confusing. The current, corrected version indicates that:

(1) The velocity on the lower and upper edges of the plate is set to 0 in the *y* direction only : the *x*-component of the velocity can be non-zero.

(2) The velocity is strictly zero only on the lower-left corner of the plate (indicated by the triangle).

(3) No confinement is applied on the lateral sides, hence these boundaries may move freely. However, as these small-deformation simulations are run for a short time (until the *cumulative* applied deformation is of at most 10% -there was an error in the text [l.4, p. 27] so we corrected it- of the size of an average, *single* model element), the position of grid nodes is *not* updated in time (hence the FE spatial discretization is defined based on the initial mesh grid, i.e., is not updated in time). The lateral boundaries therefore do not "move" in the simulations presented here. We now mention this point in section 6.

(4) As deformations are small, mass-conservation is not prescribed (see page 27, first paragraph).

p.27 l.26f: "so that to be representative:" the conjunction "so that" should not be followed by an infinitive.
OK.

p.28 l.1: drop the comma.
OK.

p.28 l.10-26: Maybe move this to the literature discussion?
Agreed, we move this to the Introduction section.

p.29 l.8: "... is its strong anisotropy" would this finding not call for the use of a tensor damage variable? Please explain why a scalar damage model is sufficient.

Again, a scalar damage model is sufficient to generate large-scale anisotropy in an elastic medium. The kernel of elastic interactions and the resulting perturbation of the stress field in such a material becomes anisotropic as soon as some spatial heterogeneity in its mechanical strength, or if the applied forcing is not purely isotropic with respect to this heterogeneity or to the geometry of the experiment (see response to major comment above).

p.30 l.15 "over undamaged parts" rather "in undamaged parts?"
OK.

p.30 l.19: "spatial distribution of the damage criteria" isn't the damage criterion the same everywhere? Only its parameters vary.

If the parameters (e.g., *C*) of the damage criterion vary, then the damage threshold itself varies. Here the local damage threshold varies with the cohesion, *C*. We replace "criteria" by "threshold" here, to make this point clearer.

p.32 l.6: "we use the output of strain rate fields from simulations..." isn't it rather the output of simulations, not that of strain rate fields?
We removed "the".

p.32 l.24: "....and clusters in space ..." This seems somehow syntactically lost in the sentence?
Unfortunately, we do not understand this comment. We replaced this formulation by "localizes again".

p.34 l.9: "of the discrete failure events:" drop the article.
OK.

p.34 l. 25: "... the of damage rate are anti-correlated" something missing here?
Yes: *increments* of the damage rate.

p.35 l.6: "mean total deformation, that's a deformation rate, not a deformation.
OK.

p.36 l.7: "permanent deformations within the material" There are no deformations outside of the material, so no need to state this.
OK.

p.36 l.9 "show the Maxwell-EB model simulates...." maybe make the beginning of the subordinate clause clear by using "that". Or even better split into two sentences.
OK.

p.36 l.26: "internal stress within the material" why not just "the Cauchy stress"? Or even "the stress?" I guess there are neither external stresses, nor stresses outside of the material.
See previous comment: "internal stress" is the term used extensively by the sea ice modelling community.

p. 37 l.1: "of the material" instead of "the material's"
OK.

p.37 l.20: "carrying numerical experiments" rather "carrying out"
OK.

p.39 l.20: typo "shear faaults"
OK.

p.50, Fig. 6b,c: Excessive use of colored plots. Except for the yellow one, they all look more or less the same to me. I am probably not the only one who will have trouble distinguishing those plots: statistically, you can expect that about one out of ten male readers has a similar color vision deficiency. So it probably makes sense to use dash patterns instead, or to add plot labels.
Agreed: we switched to black and white and added markers wherever possible.

p.51, Fig. 7a (and various other figures): Are the units dimensionless? If so, this should be made clear, e.g. with the tilde notation used in the text.
Thank you for this comment: this was indeed not made clear since we forgot to mention that we dropped the "~" notation for all adimensional variables in the Results section. We now make this point clearer by adding a mention to this effect and by rearranging the order of a few sentence in section 6 (formerly section 4).

**Answers to reviewer 2**

**GENERAL COMMENTS**

The thoroughness and detail of the manuscript are commendable, although in some places the presentation was somewhat confusing as a result. I would recommend splitting the Introduction into separate Introduction and Background/Theory sections. A shorter Introduction section that clearly outlines the motivation for the new model, the most relevant context, and the general approach would benefit the reader. As it is written, the Introduction currently is very long and detailed but in some places confusing in terms of both the writing and the relevance of this level of detail. In other places, relevant details seem to be left out, and it seems to be left to the reader to be familiar with all of the references in order to understand certain points. It seems to be an excellent review of the state of sea ice modelling, and demonstrates that the authors have a good grasp of the field, but as a reader I found myself a little "lost in the weeds" at times.

We followed your suggestion and divided the introduction into an *Introduction* and a *Background* section. In particular, we also integrated the suggestion of reviewer 1 of moving the discussion of heterogeneity, intermittency and anisotropy (at the beginning of the Results section) to the Background section. We believe this helps presenting our motivations in developing this new rheological framework more clearly, as we indeed aim to create a model that is able to reproduce these (3) all-important characteristics of sea ice deformation.

Even in a fully viscous model of ice deformation, the stress balance can be non-local (for instance, in a glaciology context, viscous ice stream or ice shelf deformation is described by a stress balance that is inherently nonlocal, (e.g. MacAyeal, 1989)). You seem to be implying in several places that "long-range" interactions must come from elastic deformation (e.g. p 4, l 14). Am I reading this wrong, or are you indeed stating that long-range interactions can only be accounted for by elastic interactions?

This is indeed not what we implied. You are totally right on the point that stress balance can be non-local in fully viscous models, in other words that viscous deformations can redistribute stresses in a non-local manner. In our case, elastic interactions by nature redistribute the stress over *long* distances and need to be accounted for in order for the heterogeneity of deformation to be adequately represented in models of the ice pack.

The results of the model are mesh-dependent, as damage localizes to the scale of an individual element. This is often viewed as a negative result, because the results of the model thus depend on how the user sets it up. However, many different approaches for nonlocal regularization of damage models have been proposed and adopted in a variety of settings (e.g. Bazant and Jirasek, 2002; Borstad and McClung, 2011). In these approaches, the stresses/strains/constitutive relation are computed by integrating over an intrinsic length scale related to the scale of heterogeneity of fracturing of the material. As long as the element size is smaller than this intrinsic length scale, the results of the model are independent of the resolution of the mesh. I think the authors should mention this type of approach in the manuscript, and discuss whether it might be feasible to produce mesh-objective results.

Thank you very much for this comment: this is a very important point that we did not discuss, mainly to keep the paper as short as possible.

Introducing an intrinsic length scale, i.e., a correlation length, $\xi$, for damage that is larger than the model grid cell would indeed allow the model solution to converge. However, in the context of sea ice modelling, introducing a correlation length $\xi > \Delta x$ would not be physical, as the scale of natural heterogeneities (thermal cracks, brine pockets, etc) within the ice cover that serves as stress concentrators is much smaller than the typical model spatial resolution (on the order of a few to several kilometres). Furthermore, invariance of sea ice fracturing, as revealed from floe sizes distributions, holds down to the meter scale (*Weiss, 2003*). Hence here, disorder is introduced at the smallest available scale: that of the mesh element (through the field of cohesion, $C$). We believe this is the most rigorous choice in terms of representing the physics behind the deformation of the ice cover, but as a result the model solution is mesh-dependent and does not converge locally.

We agree that this point deserves a more extensive discussion in the paper. Hence we somewhat reformulated lines 15-22, page 11 and 1-7, page 12 to clarify the role of the spatial noise introduced in the model through the cohesion variable, $C$, and added a few line discussing non-local damage and convergence in section 5.1 (former lines 19-27, page 30, and 1-11, page 31).

The discussion of the damage formulation is a bit confusing. You mention in the text (p 14, l 24-26) that stresses outside the failure envelope are non-physical. However, unless I am missing something, you seem to be calculating your damage variable according to the distance beyond the failure envelope. It seems, then, that damage is a sort of constitutive post-processing to "correct" the stress level such that it lies directly on the failure envelope. Some clarification is needed in the text on this point, since your schematic representation of the failure envelope in Figure 2 seems to contradict what you state in the text. Damage is based on the distance of the stress state outside of the envelope, and yet a stress state outside the envelope is non-physical...

See response to reviewer 1's comment above. The damage of an element is calculated such that, *just after* a local damage event, the stress state of the damaged element lies on the envelope.

*I'm confused as to why a separate term for the ice concentration (A) is needed, as this seems at least partially redundant with damage. Why is it necessary to have both a concentration term and a damage term that modify both the elastic modulus and the viscosity? Isn't there some redundancy here, as a damaged fault/lead will necessarily have a reduced ice concentration? The ice concentration term seems to be simply added to the model at the very end of the model discussion, without much explanation.*

We indeed did not discuss sea ice concentration in length in the paper, since the definition of the ice concentration is the same in the Maxwell-EB model as in typical continuum sea ice models (it is the ice-covered surface by unit area, hence it varies between 0 and 1). Its equation of evolution is therefore also similar to that employed in these models.

The coupling between the ice concentration and both mechanical parameters (see equations 17, 18) here was inspired from the coupling of *A* to the ice strength in compression parameter (*P*) suggested in the VP model of *Hibler* (1979) as well as from the coupling between the elastic modulus *E* and *A* suggested by *Girard et al.* (2011) and *Bouillon and Rampal* (2015).

In the case of the elastic modulus, *E*, this formulation represents the fact that when the ice concentration drops below about 90%, internal stresses become negligible and the ice is essentially in free drift. In the case of $\eta$, this dependence on ice concentration is consistent with the rapid decay of the apparent viscosity of granular media when decreasing their packing fraction from the close-packed limit (*Aranson, 2006*).

While concentration might indeed seem partially redundant with damage, we emphasize that the two variables represent different things, as the ice pack might be densely fractured, hence not withstand large stresses, but still retain a high concentration (for instance under convergent motion). Nevertheless, we agree that this parameterization, which we chose here so that to be consistent with the approach taken in previous sea ice models, could eventually be refined.

**SPECIFIC COMMENTS**

*25: "or"*
OK.

*p 4, l 6-9: how have these VP hypotheses been found inconsisent? Can you summarize these for the reader?*
The VP hypotheses have been found inconsistent in many respects, which are discussed in length in the studies cited here (*Weiss et al., 2007, Coon et al., 2007, Rampal et al, 2008*, see lines 8-9, page 3). Although discussing these inconsistencies could be pertinent here, it would also make the introduction significantly longer. Hence we chose to add only a short list of the main points at the end of this sentence (line 9, page 3) with the references to the relevant studies.

*check English spelling throughout the document, a number of words are misspelled (looses instead of loses, euqation, it's instead of its, dependance, assymptotes,...)*
OK.

*p 10, l 20: there is some inconsistency in the text formatting of the different versions of "I" for the identity tensors*
We reformulated the definition of **K** using index notation.

*p 13, l 22-23: I was confused about what h is here*
Thank you for catching this: *h* refers to the ice thickness and its definition in now defined.

*p 19, l 18: might there be other contexts or model setups (e.g. realistic domains) for which the minimum value of d might come into play? The results of a damage model can be quite sensitive to this choice.*
Problems, both conceptual and numerical, arise when the value of *d is* zero, which according to the evolution equation for *d* (line 20, p. 15 or equation 9) occurs if, and only if, the initial value of *d* ($d^0$) is set to zero. In the uniaxial experiment presented here, this problem does not arise since the experiment is started from a homogeneous, undamaged state ($d^0 =$ 1). A regularization technique, such as the one presented in this paragraph, can be used to avoid numerical problems when $d^0$ is allowed to take the value of 0. Alternatively, a cutoff minimum value for *d* could be used.

As this section brings essentially useless complexity in the context of the simulations presented in this paper, we decided to remove it in the revised manuscript (see response to reviewer 1's comments).

p 20, l 11-13: Some motivation or explanation is needed for why you choose to write the momentum equation in terms of internal stress rather than the vertically integrated stress tensor, especially if you are departing from what is more commonly done in the sea ice modelling community.

We chose to incorporate the vertical integration of the internal in-plane stress in the momentum equation instead of defining $\boldsymbol{\sigma}$ as the vertically-integrated stress tensor to avoid confusion when evaluating the distance to the damage criterion, i.e., when comparing the local state of stress $\boldsymbol{\sigma}$ to the critical tensile stress $\boldsymbol{\sigma_t}$ and critical stress with respect to the Mohr-Coulomb criterion $\boldsymbol{\sigma_c}$, both defined in Pascals ($Nm^{-2}$) as the cohesion $C$ of the material. Defining $\boldsymbol{\sigma}$ as the vertically-integrated stress tensor would indeed necessitate redefining $\boldsymbol{\sigma_t}$ and $\boldsymbol{\sigma_c}$ in terms of an *effective* cohesion, $C \times h$. This approach is taken in the recently developed NeXtSIM model, based on the Elasto-Brittle rheology of Girard et al. (2011), and as this point was explained in the paper by Bouillon et al. (2015), we do not feel it needs to be discussed in length here, but we do agree that a clear reference to this paper is needed. Hence we reformulate the lines 10 to 13 as follows to explain this point more clearly:

"We assume the internal stress to be homogeneously distributed over the depth $h$ and write the momentum equation in terms of the internal stress rather than the vertically integrated stress tensor more commonly used in the sea ice modelling community. This approach was also taken in the Elasto-Brittle model of Bouillon et al. (*NeXtSIM, 2015*), as it allows a direct comparison between the local state of stress and the critical stress (here $\boldsymbol{\sigma_t}$ or $\boldsymbol{\sigma_c}$) when estimating the distance to the damage criterion."

p 23, l 6-9: why not perform a sensitivity analysis as you describe then?

We did perform sensitivity analyses on the relative values of $t_d$ and $\Delta t$ in the Maxwell-EB model. The results of such analyses however, depends on the applied forcing, domain geometry, etc, and hence is specific to each numerical experiment. Presenting the details of these analyses is therefore beyond the scope of the present paper. The remark of lines 6-9 was instead meant as a warning one should be careful and perhaps carry sensitivity analyses when using a model time step that is larger than the prescribed time of propagation of damage ($t_d$). To ensure numerical stability and allow for highest resolution of the elastic interactions in the Maxwell-EB simulations presented in this paper, we set $t_d = \Delta t$. As this also ensures the one to one correspondence between the progressive damage mechanism as described by the recursive relation of line 20, page 15 and the continuous form of equation 20, this appears to be the most logical choice. Hence to avoid giving the impression that $\Delta t$ could be set arbitrarily larger than $t_d$ in the model (at the cost of a sensitivity analysis) without any loss of physical rigor, we removed this sentence and slightly reformulated this paragraph.

p 26, l 5: you previously described the inertial term as being negligible, so why is it here? some clarification is needed.

We replace "inertial term" by the "time derivative" of the stress tensor.

The first part of the Results section is not really results, but background.

It is now included in the background section.

p 29, l 23-24: this would be helpful to state also in the figure caption

OK.

p 30, l 26-27: well, the localization scale is the element scale, so the choice of resolution dictates the localization of damage

We totally agree with this remark, and this is what we mean by "there is no *physical* scale associated with the localization of damage. Through elastic interactions, damage and deformation tend to localize at the finest scale (the mesh element)".

p 31, l 6-11: but you didn't introduce disorder initially, so you cannot claim this here.

Perhaps we do not fully understand this comment, but we indeed introduced disorder (in the damage threshold) through the field of cohesion, set at the beginning of each simulation. We try rephrasing this line as "the initial disorder introduced in the model".

p 31, l 27: "...has already been investigated in depth..." is another example of the reliance on the reader to be familiar with all of the literature you are citing. It would be more helpful to summarize the findings. What did these investigations find?

Thank you for this comment. Here we meant that damage models based on a linear-elastic constitutive law and without healing have been demonstrated to reproduce a highly heterogeneous deformation (the finding that is most relevant for our purpose). However, as these frameworks neither include a healing mechanism nor a slow relaxation of elastic stresses, the post macro-failure behaviour of these models is physically inconsistent, and only the path to the first rupture was analyzed. Here we seek to establish if the Maxwell-EB model, based on a viscoelastic constitutive law, has the same capability of reproducing a highly heterogeneous deformation in a partially damaged material (over subsequent healing-fracturing cycles). We hence reformulate these sentences to make this point clearer.

Yes, "increments", thank your for catching this.

Yes, resolution is increasing from top to bottom on this figure. However, you might get the impression that the resolution is the same between the experiments, especially at the beginning of the simulation (panels 1 and 2,) because the initial field of cohesion is the same in all simulations. That is, we use the field of $C$ prescribed in the $N = 10$ simulation (lowest resolution, top row). This field is interpolated onto the finer resolution grids (see lines 16 to 19, p. 29) in the other three simulations. Perhaps this point was not made clear enough, hence we slightly reformulate the description of these experiments, at the beginning of section 5.1.
As for the right $y$-axis on panel (a), thank your for catching this.

Indeed, this figure shows one realization of the experiment. However, repeating the same scaling analysis for other realizations of the experiment gave very similar results. In this sense, averaging over several realizations did not give more insight.
This is what we meant by "Repeating the procedure for subsequent healing and damaging cycles and for multiple realizations of the experiment initialized with different cohesion fields showed a similar evolution of the rate of decrease of $\langle \dot{\varepsilon}_{tot} \rangle_l$ with $l$ between macro-ruptures events, with values of β in the vicinity of the rupture consistent with previous EB model analyses (e.g., β = 0.15 ± 0.02 ; Girard et al., 2010a)."
The figure below shows for instance the result of averaging $\langle \dot{\varepsilon}_{tot} \rangle_l$ computed as a function of the spatial scale, at five equidistant stages (as indicated on figure 7a, formerly figure 6a) between the minimum in macroscopic stress that follows the propagation of a fault (red) and the maximum that precedes the next macro-rupture (purple), over the second cycle of stress build-up and macro-rupture for 5 different realizations of the uniaxial compression experiment with a resolution of $N = 100$. The total deformation rate still shows a clear power law decrease with increasing spatial scale, with β largest for the post-macro-rupture (red) and pre-macro-rupture (purple) stages. The averaging could alternatively be done over multiple stress build-up/macro-rupture cycles of a single experiment. However, we believe that giving an appropriate description of the method for partitioning of the results in different stress build-up/macro-rupture cycles and averaging $\langle \dot{\varepsilon}_{tot} \rangle_l$ over these multiple cycles so that to present an "average" figure such as the one below instead of figure 7 would make the reading of this part of the *Results* section more dense than insightful.

[Figure]

Moreover, we do not believe that *varying* the resolution would be relevant here. Increasing the resolution could indeed be interesting, in the sense that it would allow performing the scaling analysis over a larger range of space scales. Obviously it would also be computationally more expensive. Here, the analysis and the power law obtained already spans almost 2 orders of magnitude in $l$.

**REFERENCES**

Aranson, I. S. and Tsimring, L. S.: Patterns and collective behavior in granular media: Theoretical concepts, *Rev. Mod. Phys.*, 78, 641–692, doi:10.1103/RevModPhys.78.641, 2006. 8

Amitrano, D., J.-R. Grasso, and D. Hantz, 1999: From diffuse to localised damage through elastic interaction. *Geophysical Research Letters*, 26, 2109–2112.

Bažant, Z.P., Jirásek, M., 2002. Nonlocal integral formulations of plasticity and damage: survey of progress. *Journal of Engineering Mechanics,* 128 (11), 1119–1149.

Bouillon, S., and P. Rampal, 2015: Presentation of the dynamical core of neXtSIM a new sea ice model. *Ocean Modelling*, 91(0), 23–37.

Coon, M., Kwok, R., Levy, G., Puis, M., Schreyer, H., and Sulsky, D.: Arctic Ice Dynamics Joint Experiment (AIDJEX) assumptions revisited and found inadequate, *J. Geophys. Res.*, 112, C11S90, doi:10.1029/2005JC003393, 2007. 3, 13, 36

Cowie, P. A., C. Vanneste, and D. Sornette, 1993: Statistical physics model for the spatiotemporal evolution of faults. *Journal of Geophysical Research*, 98(B12), 21,809–21,821.

Eshelby, J. D., 1957: The determination of the elastic field of an ellipsoidal inclusion, and related problems. *Proceedings of the Royal Society of London A: Mathematical, Physical and Engineering Sciences,* 241(1226), 376–396.

Girard, L., J. Weiss, J. M. Molines, B. Barnier, and S. Bouillon, 2009: Evaluation of high-resolution sea ice models on the basis of statistical and scaling properties of arctic sea ice drift and deformation. *Journal of Geophysical Research*, 114(C08015).

Girard, L., Amitrano, D., and Weiss, J.: Failure as a critical phenomenon in a progressive damage model, *J. Stat. Mech.-Theory E.*, P01013, doi:10.1088/1742-5468/2010/01/P01013, 2010a. 5, 14, 16, 29, 30, 31, 32, 33

Girard, L., S. Bouillon, J. Weiss, D. Amitrano, T. Fichefet, and V. Legat, 2010b: A new modeling framework for sea ice models based on elasto-brittle rheology. *Annals of Glaciology,* 52(57).

Hibler, W. D. I., 1979: A dynamic thermodynamic sea ice model. *Journal of Physical Oceanography,* 9(7), 815–846.

Krug, J., Weiss, J., Gagliardini, O., and Durand, G.: Combining damage and fracture mechanics to model calving, *The Cryosphere*, 8, 2101–2117, doi:10.5194/tc-8-2101-2014, http://www.the-cryosphere.net/8/2101/ 2014/, 2014.

Keller, A. and Hutter, K.: A viscoelastic damage model for polycrystalline ice, inspired by Weibull-distributed fiber bundle models. Part I: Constitutive models, *Continuum Mechanics and Thermodynamics*, 26, 879–894, doi:10.1007/s00161-014-0348-7, http://dx.doi.org/10.1007/s00161-014-0348-7, 2014.

Main, I., 1996: Statistical physics, seismogenesis, and seismic hazard. *Reviews of Geophysics*, 34(4), 433–462.

Rampal, P., Weiss, J., Marsan, D., Lindsay, R., and Stern, H.: Scaling properties of sea ice deformation from buoy dispersion analysis, *J. Geophys. Res.*, 113, C03002, doi:10.1029/2007JC004143, 2008. 3, 4, 28, 35

Weiss, J.: Scaling of Fracture and Faulting of Ice on Earth, *Surveys in Geophysics,* 24, 185–227, doi:10.1023/A:1023293117309, http://dx.doi.org/10.1023/A:1023293117309, 2003

Weiss, J., 2008: Intermittency of principal stress directions within Arctic sea ice. *Physical Review E*, 77(5), 056,106.

Weiss, J., 2013: Drift, Deformation, and Fracture of Sea Ice, A perspective across scales. *SringerBriefs in Earth Sciences*, Springer Netherlands.